# Identifying Neural Dynamics Using Interventional State Space Models

**Amin Nejatbakhsh** [1]   **Yixin Wang** [2]

## Abstract

Neural circuits produce signals that are complex and nonlinear. To facilitate the understanding of neural dynamics, a popular approach is to fit state space models (SSM) to the data and analyze the dynamics of the low-dimensional latent variables. Despite the power of SSM to explain the dynamics of neural circuits, these models have been shown to merely capture statistical associations in the data and cannot be causally interpreted. Therefore, an important research problem is to build models that can predict neural dynamics under causal manipulations. Here, we propose interventional state-space models (iSSM), a class of causal models that can predict neural responses to novel perturbations. We draw on recent advances in causal dynamical systems and present theoretical results for the identifiability of iSSM. In simulations of the motor cortex, we show that iSSM can recover the true latents and the underlying dynamics. In addition, we illustrate two applications of iSSM in biological datasets. First, we applied iSSM to a dataset of calcium recordings from ALM neurons in mice during photostimulation. Second, we applied iSSM to a dataset of electrophysiological recordings from macaque dlPFC during micro-stimulation. In both cases, we show that iSSM outperforms SSM and results in identifiable parameters. The code is available at `https://github.com/amin-nejat/issm`.

## 1. Introduction

Understanding neural data requires identifying the dynamics that underlie it. The principled way to achieve this is through causal perturbations. When a perturbation is delivered, the activity of perturbed neurons is dissociated from their upstream neurons, facilitating inspection of the circuit dynamics when certain edges are functionally removed from the circuit. This powerful strategy enables testing sophisticated neural hypotheses. For example, O'Shea et al. (2022) used perturbations to understand whether dynamics in the motor cortex are path-following (driven by an upstream brain region), low-dimensional, or high-dimensional. Another example by Feulner et al. (2022) uses a similar strategy to investigate whether feedback drives plasticity for rapid learning in the motor cortex. Another study by Sanzeni et al. (2023) uses optogenetic perturbations to uncover the degree of coupling in the visual cortex of mice and monkeys. They demonstrate through modeling that under strong network coupling, perturbations lead to a reshuffling of responses within the circuit.

The main insight of these works is that in the absence of perturbations (i.e. *observational regime*), neural dynamics are confined to low-dimensional spaces, and models that are built upon observational data are not able to capture neural dynamics outside of the low-dimensional space. However, during perturbations (i.e. *interventional regime*), the neural state is driven outside of the task space, providing more information about dynamics in the global neural state space (Jazayeri & Afraz, 2017). This insight allows us to build sophisticated hypotheses that can only be tested using perturbations (Fig. 1). Interventional studies are critical for determining the causal contribution of neural dynamics to behavior and perception. For example, a study by Shahbazi et al. (2022) used targeted electrical stimulation to manipulate a monkey's perception.

Furthermore, many of the popular models used in neuroscience suffer from identifiability issues (Maheswaranathan et al., 2019). In one line of work, researchers have developed similarity metrics that are agnostic to non-identifiability transformations (see Sucholutsky et al. (2023) for a review). However, in many cases, the model parameters or latent variables are biologically meaningful, and recovering them is desired. Therefore, the need for developing identifiable models for neuroscience data analysis is an overarching goal. For example, Zhou & Wei (2020) used an identifiable VAE as opposed to a vanilla VAE and inferred latent variables that encode the geometry of the task in an unsupervised manner.

---

[1]Center for Computation Neuroscience, Flatiron Institute, New York, US [2]University of Michigan, Michigan, US. Correspondence to: Amin Nejatbakhsh <anejatbakhsh@flatironinstitute.org>.

*Proceedings of the 42nd International Conference on Machine Learning*, Vancouver, Canada. PMLR 267, 2025. Copyright 2025 by the author(s).

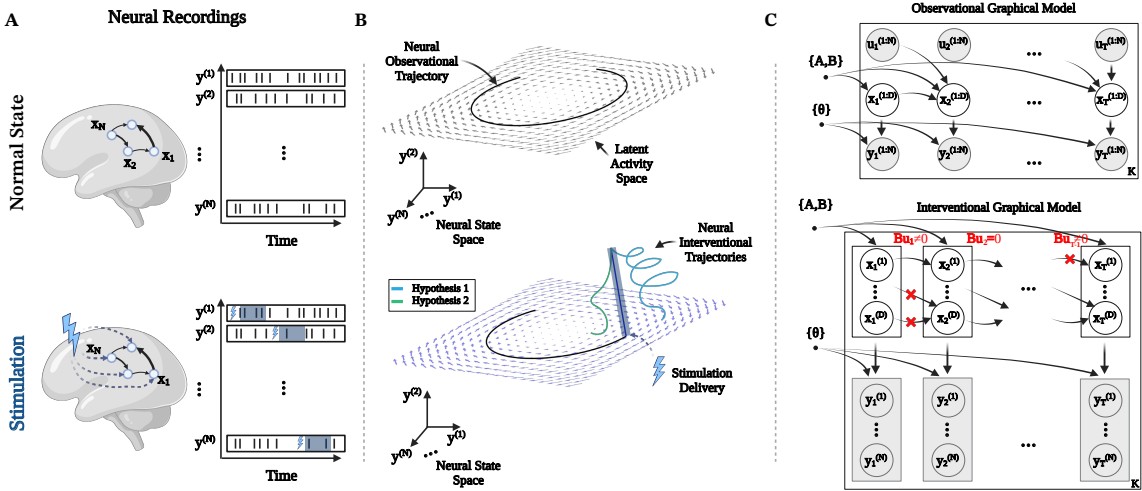

Figure 1. *Overview.* (A) Schematic of neural recordings in observational (top) and interventional (bottom) regimes. (B) Neural dynamics in the state space. The observational data (top) is confined to a low-dimensional task space, whereas the interventional data (bottom) explores the state space, enabling the testing of causal neural hypotheses. (C) Graphical model of SSM (top) compared with iSSM (bottom).

## 2. Related Work

**State space models** To understand neural circuits, a popular strategy is to build low-dimensional state space models (SSM). Driven by the neural manifold hypothesis, we often assume that neural data lies on a low-dimensional manifold. The challenge then becomes discovering the latent manifold and characterizing how the dynamics evolve in the low-dimensional space. Subsequently, a suite of SSMs has been developed covering a wide range of assumptions and applications. A typical SSM follows a dynamic model and an emission model described by the following equations:

$$\boldsymbol{x}_{t+1} = g_{\boldsymbol{\theta}}(\boldsymbol{x}_t) + \boldsymbol{\epsilon}_t, \quad \boldsymbol{y}_t = f_{\boldsymbol{\theta}}(\boldsymbol{x}_t) + \boldsymbol{\delta}_t, \quad (1)$$

$$\boldsymbol{\epsilon}_t \sim p(\boldsymbol{\epsilon}_t), \quad \boldsymbol{\delta}_t \sim p(\boldsymbol{\delta}_t). \quad (2)$$

With this general formulation, models depart based on the specification of $g_{\boldsymbol{\theta}}, f_{\boldsymbol{\theta}}, p(\boldsymbol{\epsilon}_t), p(\boldsymbol{\delta}_t)$. Linear dynamical systems (LDS) assume that both $g_{\boldsymbol{\theta}}, f_{\boldsymbol{\theta}}$ are linear and $p(\boldsymbol{\epsilon}_t), p(\boldsymbol{\delta}_t)$ are multivariate normal distributions. A separate line of work assumes that $g$ is switching linear and develops algorithms that jointly infer switching times as well as latent states (Petreska et al., 2011; Linderman et al., 2017; Fox et al., 2008). These models have been successful, in particular when there are abrupt changes in the dynamics.

Although SSMs have been primarily used for fitting observational data, there have been a few attempts to apply them to interventional data as well. However, responses to perturbations are often modeled as additive, which makes the SSM models non-causal. We will elaborate on this further in section 3.1.

**Model identification in static data** The emerging field of causal representation learning provides statistical treatments for recovering the true parameters of statistical models (Schölkopf et al., 2021). Most of the developments correspond to static models and can be broadly categorized into identification using observational or interventional data. **(1) Observational:** While early theoretical guarantees have been limited to linear mixing and asymmetric noise (Comon, 1994), these results have been extended to nonlinear mixing (Locatello et al., 2019; Xi & Bloem-Reddy, 2023), and nonlinear mixing with observation noise (e.g., VAEs) (Khemakhem et al., 2020a), and multi-environment data (Lachapelle et al., 2023). **(2) Interventional:** With access to interventional data, identifiability results can be extended to broader classes of models. Lippe et al. (2022) shows that with sparse interventions, we can recover latents up to permutation, scaling, and offset. Ahuja et al. (2023) utilize independent support properties (Wang & Jordan, 2021) and provide identifiability guarantees. These results have been further extended to nonparametric latents with linear and nonlinear mixing functions (von Kügelgen et al., 2023; Buchholz et al., 2023; Varici et al., 2023).

**Model identification in dynamic data** More recently, theoretical results on statistical model identification have been extended to Markov models and switching linear dynamical systems (Balsells-Rodas et al., 2023). These results provide the identification of the model parameters up to a class of nuisance transformations (e.g., affine). Most relevant to our work are Yao et al. (2022; 2021). The main shortcoming of these works is that they do not incorporate noise in the observation space, which is crucial for modeling biological

datasets. Previous work can be broadly categorized into two groups. Some studies consider the transient interventional effects, while others investigate the persistent effects in the stationary regime (Schölkopf & von Kügelgen, 2022; Malinsky & Spirtes, 2018; Besserve & Schölkopf, 2022; Benkő et al., 2018; Malinsky & Spirtes, 2018; Peters et al., 2022). Hansen & Sokol (2014) uses differential equations as structural equations in dynamical systems. Ahuja et al. (2021) considers (deterministic) linear dynamics (referred to as mechanism) and nonlinear emissions (referred to as rendering) and proves that the latent space of such a model is identifiable from observational data up to mechanism invariances. Lippe et al. (2023) show that for linear dynamics, if we have access to binary interventional data, then the latents are identifiable up to permutation. Yao et al. (2022); Song et al. (2024b) focus on the identification of latent non-stationary dynamics using observational data. Hyvarinen & Morioka (2016; 2017); Hyvarinen et al. (2019); Hälvä et al. (2021) focus on extending nonlinear ICA and its identifiability results to temporal settings. They impose constraints on the mixing function and the latent dynamics to achieve identification using only observational data.

In addition to the statistical literature on model identification, recent work in dynamical systems theory has utilized the Koopman theory to find conditions, such as sampling frequency, for the exact identification of the continuous-time dynamical systems from sampled data (Zeng et al., 2022).

## 3. Methods

Here, we develop a new class of latent variable models that aim to capture neural dynamics in both observational and interventional regimes. We base our model on the framework of Causal Inference (CI) (Pearl et al., 2016). Instead of directly modeling the joint distribution of the data, CI uses structural equations to describe the generative process of the data. In this framework, interventions are modeled as changing the structural equations. Equivalently, when an intervention is performed on a node, it is disconnected from all its parents in the generative model, and its distribution is set to a new distribution. A major benefit of modeling the interventions in this way is that having access to interventional data allows us to identify the model parameters, as we will see in section 3.3.

Although our framework is general and can be applied to various types of SSMs, in this paper, we focus on adding interventional components to the SSMs with linear dynamics and nonlinear observations. These models are natural extensions of linear models with a higher capacity to express complex datasets (Gao et al., 2016). It has been argued theoretically that linear dynamics in a latent space with sufficiently large dimension, followed by a nonlinear emission, is powerful to model any dynamical system (Koopman,

1931). Therefore, the model we consider here theoretically has the capacity to fit complex datasets.

### 3.1. Interventional State Space Models

**Notation** Consider an experiment with $N$ recorded neurons over $T$ time steps repeated for $K$ trials. We denote neural responses at time $t$ by $\boldsymbol{y}_t$, where $\boldsymbol{y}_t$ is an $N$-vector that concatenates the spike counts or calcium activities of all neurons. We assume the existence of a time-dependent latent variable $\boldsymbol{x}_t \in \mathbb{R}^D$ where $D$ is the dimension of the latent space. We present the interventional model and elaborate on its differences from the observational model.

The first modeling assumption that distinguishes iSSM from SSM is that we assume perturbing neurons affect the latent dynamics directly, which will consequently affect neural responses in the next time point according to the emissions model. The second, more critical assumption is that whenever a latent node is perturbed, its activity is dissociated from all its upstream nodes. This assumption is easy to incorporate in a linear model, which is achieved by ignoring the columns in the dynamics matrix corresponding to the perturbed node. Denoting the interventional input to individual channels at time $t$ by $\boldsymbol{u}_t \in \mathbb{R}^M$, we model $\boldsymbol{x}, \boldsymbol{y}, \boldsymbol{u}$ as:

$$\boldsymbol{x}_{t+1} = 1\{\boldsymbol{B}\boldsymbol{u}_t = 0\} \otimes \boldsymbol{A}\boldsymbol{x}_t + \boldsymbol{B}\boldsymbol{u}_t + \boldsymbol{\epsilon}_t, \quad (3)$$

$$\boldsymbol{y}_t \sim P(\boldsymbol{y}_t | f_{\boldsymbol{\theta}}(\boldsymbol{x}_t)). \quad (4)$$

where $\boldsymbol{\epsilon}_t \sim \mathcal{N}(\boldsymbol{0}, \boldsymbol{Q})$ and $\otimes$ denotes element-wise multiplication. $\boldsymbol{A} \in \mathbb{R}^{D \times D}$ captures latent dynamics, while $\boldsymbol{B} \in \mathbb{R}^{D \times M}$ captures the effect of neural perturbations on latent dynamics. We place a centered Laplace prior on $\boldsymbol{B}$ with the scale parameter $s$ to encourage its sparsity, which is critical for the identifiability of the results as we discuss in section 3.3. $\boldsymbol{Q} \in \mathbb{R}^{D \times D}$ is the covariance of $\boldsymbol{\epsilon}_t$ and $f_{\boldsymbol{\theta}}$ is a generic nonlinear function mapping latents to observations. If the intervention $\boldsymbol{u}_t$ is zero, the model follows observational dynamics, but in the presence of an intervention, the model decouples the intervened node from its parents. We term this model interventional SSM or iSSM. In section 3.3 we theoretically characterize the conditions under which iSSM is identifiable.

**Justification of intervening on latents:** Many causal experiments in neuroscience (both optical and electrophysiological) still do not perform interventions at the single-neuron level. In some cases, the light-gated proteins (e.g., Channelrhodopsin) are expressed in all the neurons of a certain subtype, and broad-field illumination is used to activate the population of neurons of that subtype. In other experiments, light-gated proteins are expressed in a sparse subset of neurons, and two-photon lasers are used to activate only those neurons. In these cases, due to the spillover effect of lasers, it is nearly impossible to activate individual neurons with

enough spatial precision. Similarly, for electrophysiological techniques such as micro-stimulation, each electrode targets several neurons located in its vicinity as opposed to a single neuron. While iSSM allows for performing interventions on individual neurons by setting $\boldsymbol{B}$ to the identity matrix (see Supp. Fig. 6 for an example of this), it is still useful to think of latents as providing a natural grouping of the neurons into behaviorally relevant subsets. We demonstrate this in our results in section 4.

## 3.2. Inference

Since our model involves a nonlinear emission as well as a non-conjugate noise model, we resort to variational inference techniques. Our goal is to infer the posterior distribution $p_{\boldsymbol{\Theta}}(\boldsymbol{x}_{1:T}|\boldsymbol{y}_{1:T}, \boldsymbol{u}_{1:T})$ while optimizing the parameters $\boldsymbol{\Theta}$. We follow the methodology of reparameterization and amortized inference, but adapt some parts to our specific interventional scheme. For a review of variational methods for state space models see Archer et al. (2015). Denoting the approximate posterior distribution by $q_{\boldsymbol{\Phi}}(\boldsymbol{x}_{1:T})$, the ELBO loss function is presented below:

$$\mathcal{L}(\boldsymbol{\Phi}, \boldsymbol{\Theta}) = \mathbb{E}_{\boldsymbol{\eta} \sim \mathcal{N}(\mathbf{0}, \boldsymbol{I})} \left[ \log p_{\boldsymbol{\Theta}}(\boldsymbol{y}_{1:T}, \boldsymbol{u}_{1:T}, \boldsymbol{x}_{1:T}) \right.$$
$$\left. - \log q_{\boldsymbol{\Phi}}(\boldsymbol{x}_{1:T}|\boldsymbol{y}_{1:T}, \boldsymbol{u}_{1:T}) \right]$$

where $\boldsymbol{x}$ is reparameterized as $\boldsymbol{x}(\boldsymbol{\eta}) = \boldsymbol{\mu}_{\boldsymbol{\Phi}} + \boldsymbol{\sigma}_{\boldsymbol{\Phi}} \boldsymbol{\eta}$. The functions $\boldsymbol{\mu}, \boldsymbol{\sigma}$ are typically parameterized by neural networks (called the recognition network) with an architecture that matches the dataset domain. Here, we choose an LSTM for the recognition network.

Another important addition that makes the inference in our model possible is to apply the interventional structure directly to the approximate posterior during training. To do this, we replace $\boldsymbol{\mu}_t$ with $\mathbb{1}\{\boldsymbol{B}\boldsymbol{u}_t = 0\} \otimes \boldsymbol{\mu}_t + \boldsymbol{B}\boldsymbol{u}_t$ during each iteration of optimization. This ensures that the interventional data indeed manipulates the causal graph consistently in the approximate posterior. We refer the reader to the supplementary C.1 for more details on inference.

## 3.3. Theoretical results: On the identifiability of iSSM

We provide sufficient conditions for the identifiability of iSSMs. We show that, given a sufficient set of *do*-interventions, one can identify both the latent dynamics matrix $\boldsymbol{A}$ and the mixing function $f_{\boldsymbol{\theta}}(\cdot)$ of the iSSM. This identifiability of the latent dynamics enables us to extrapolate to novel, unseen interventions.

To identify the latent dynamics of iSSM, we proceed in three steps: (1) identify $P(\{f_{\boldsymbol{\theta}}(\boldsymbol{x}_t)\}_{t \in T})$ from the observed data distribution $P(\{\boldsymbol{y}_t\}_{t \in T})$; (2) identify $f_{\boldsymbol{\theta}}$ and $P(\{\boldsymbol{x}_t\}_{t \in T})$ from $P(\{f_{\boldsymbol{\theta}}(\boldsymbol{x}_t)\}_{t \in T})$ up to affine transforma-

tions; (3) further identify $f_{\boldsymbol{\theta}}$ and $P(\{\boldsymbol{x}_t\}_{t \in T})$ up to permutation, coordinate-wise shifting and scaling.

Begin with the first step of identifying $P(\{f_{\boldsymbol{\theta}}(\boldsymbol{x}_t)\}_{t \in T})$ from $P(\{\boldsymbol{y}_t\}_{t \in T})$. We make the following assumptions on the observation model.

**Assumption 3.1** (Bounded completeness of $P(\boldsymbol{y}_t|\boldsymbol{z}_t)$)**.** The function $P(\boldsymbol{y}_t|\boldsymbol{z}_t)$—where $\boldsymbol{z}_t = f_{\boldsymbol{\theta}}(\boldsymbol{x}_t)$—is bounded complete in $\boldsymbol{y}_t$. Specifically, a function $f(X, Y)$ is bounded complete in $Y$ if $\int g(X)f(X, Y)\mathrm{d}X = 0$ implies $g(X) = 0$ almost surely for any measurable function $g(X)$ bounded in $L_1$-metric (Yang et al., 2017).

When the observational model satisfies the bounded completeness assumption, we can identify $P(\{f_{\boldsymbol{\theta}}(\boldsymbol{x}_t)\}_{t \in T})$ from $P(\{\boldsymbol{y}_t\}_{t \in T})$. (We detail the proof in Appendix C.) Many common functions $P(\boldsymbol{y}_t|\boldsymbol{z}_t)$ satisfy the bounded completeness condition, including exponential families (Newey & Powell, 2003), location-scale families (Hu & Shiu, 2018), and nonparametric regression models (Darolles et al., 2011). It is a common assumption to guarantee the existence and the uniqueness of solutions to integral equations, most commonly used in nonparametric causal identification in proxy variables and instrumental variables (Miao et al., 2018; Yang et al., 2017; D'Haultfoeuille, 2011). We refer the readers to Chen et al. (2014) for a detailed discussion of completeness.

We next proceed to identifying $f_{\boldsymbol{\theta}}$ and $P(\{\boldsymbol{x}_t\}_{t \in T})$ up to affine transformations. We require the following assumption on the mixing function $f_{\boldsymbol{\theta}}$.

**Assumption 3.2** (Mixing function)**.** The mixing function $f_{\boldsymbol{\theta}}(\cdot)$ is piecewise linear, continuous, and injective.

While the piecewise linear assumption may appear restrictive, we note that it entails flexible choices of $f_{\boldsymbol{\theta}}(\cdot)$, including (deep) ReLU networks that can approximate complicated functions. We finally leverage the interventional data to achieve coordinate-wise identification of $f_{\boldsymbol{\theta}}$ and $P(\{\boldsymbol{x}_t\}_{t \in T})$. We make the following assumptions on the latent dynamics.

**Assumption 3.3** (Faithfulness)**.** When the system is unperturbed, there does not exist non-zero vectors $V, W$ such that $V^{\top}\boldsymbol{x}_{t+1} \perp W^{\top}\boldsymbol{x}_t, \forall t$.

Loosely, this assumption guarantees that the latent variables $\boldsymbol{x}_t$ are all non-trivially connected. In other words, there does not exist linear transformations such that the two neighboring timestep $V^{\top}\boldsymbol{x}_{t+1}$ and $W^{\top}\boldsymbol{x}_t$ will become independent. This assumption guarantees that any independence relationships we observe in the dynamical system is due to the intervention; it will facilitate the alignment of latent variables. We further describe the requirements of the interventions that need to be performed for identifying iSSM. Under these assumptions, we can achieve the identification of iSSM as follows.

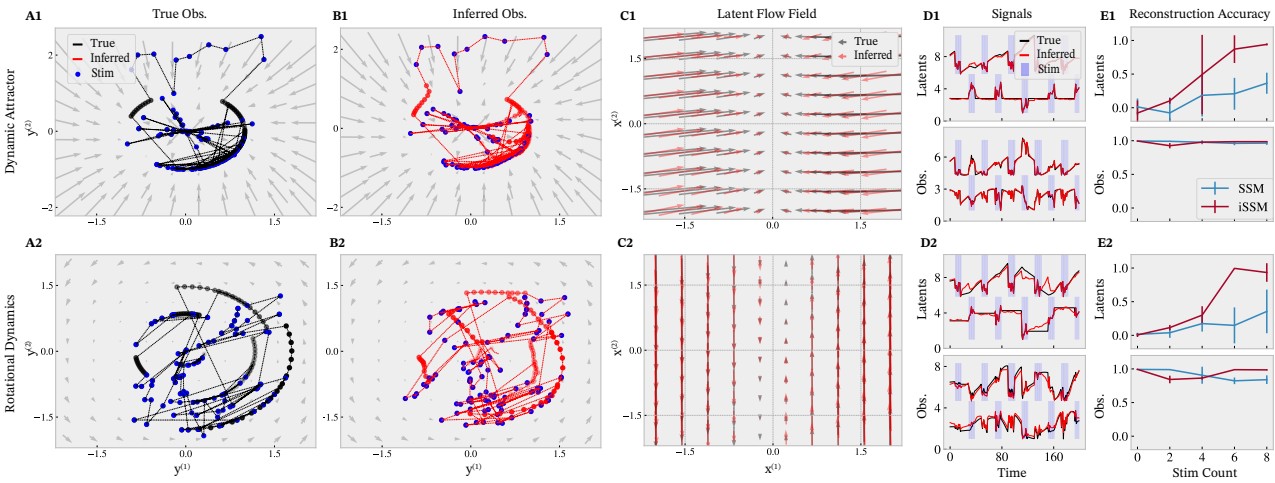

*Figure 2. Models of Motor Dynamics.* (A1,A2) Flow field underlying dynamic attractor (A1) and rotational dynamics (A2) models of the motor cortex and sampled interventional trajectories. Blue dots represent stimulation times. (B1,B2) Reconstruction of the data using iSSM. (C1,C2) True (black) and inferred (red) linear latent flow fields. The true fields are generated from the evolution of $\boldsymbol{x}_t$ in equations 5 and 6. The inferred flow fields are generated according to the fitted latent linear dynamics $\boldsymbol{A}$. (D1,D2) True (black) and inferred (red) dynamics of the latents $\boldsymbol{x}_t$ (top panels) and observations $\boldsymbol{y}_t$ (bottom panels) using iSSM. Blue shades in each plot correspond to stimulation times. (E1,E2) Comparison between SSM (observational model) and iSSM (ours). Reconstruction correlation between true and inferred latents (top panels) and observations (bottom panels) with an increasing number of interventions is shown. With more interventions, iSSM can better identify the latents.

**Theorem 3.4** (Block identifiability of iSSM and generalization to unseen interventions)**.** *Under Assumptions 3.1 to 3.3, the latent dynamics $\boldsymbol{A}$ and the mixing function of $f_{\boldsymbol{\theta}}(\cdot)$ can be block identified up to permutation, and coordinate-wise shifting and scaling; the block corresponds to a separation between intervened and unintervened latents. That is, $\hat{\boldsymbol{A}} = \boldsymbol{A}\Lambda\Pi + \boldsymbol{c}$, where $\Lambda$ is an invertible block diagonal matrix (with two blocks of size $\boldsymbol{1}^\top\boldsymbol{1}\{\boldsymbol{Bu}_t = 0\}$ and $\boldsymbol{1}^\top\boldsymbol{1}\{\boldsymbol{Bu}_t \neq 0\}$), $\Pi$ is a permutation matrix, and $\boldsymbol{c}$ is a constant vector. The observations' distribution $P(\{\boldsymbol{y}_t\}_{t\in T})$ under a novel unseen $\boldsymbol{u}_t^{new}$ intervention is also identifiable when the intervention only acts on latents in separate blocks, i.e. $\{\Lambda\Pi\}_{ij} = 0$ for any $i,j \in \{k{:}\{\boldsymbol{Bu}_t^{new}\}_k \neq 0\}$.*

The proof of Theorem 3.4 is in Appendix C. This result establishes the block identifiability of iSSM and its predictive power for unseen interventions. It says, given a single intervention trial, one can separate out the intervened latents from the unintervened ones through iSSM. It will also enable us to extrapolate to novel, unseen interventions as long as the novel interventions only touch upon latents that are already separated. The key idea of the proof relies on independence testing among the latents, built on the observation that intervention would make the intervened latent independent of the values of all other latents in previous time steps. This result illustrates how interventions can help identify latent variables by inducing statistical independence among the latents, revealing latent dynamics in non-linear state-space models.

Below we extend Theorem 3.4 to multiple interventions, for which we can achieve full identifiability if the interventions are sufficiently diverse.

**Assumption 3.5** (Unordered pairs condition (Hyttinen et al., 2013))**.** A set of interventions satisfies the unordered pair condition on the latents, if for any unordered pair $\boldsymbol{x}_{t,i}, \boldsymbol{x}_{t,j}$, there exists an intervention such that $\boldsymbol{x}_{t,i}$ is intervened on but $\boldsymbol{x}_{t,j}$ is not, or $\boldsymbol{x}_{t,j}$ is intervened on but $\boldsymbol{x}_{t,i}$.

**Corollary 3.6** (Identifiability of iSSM under sufficiently diverse interventions)**.** *If the interventions satisfy Assumption 3.5, then the iSSM is identifiable up to permutation, along with coordinate-wise scaling and shifting, $\hat{\boldsymbol{A}} = \boldsymbol{A}\Lambda^*\Pi + \boldsymbol{c}$ for some diagonal $\Lambda^*$. The observations' distribution under any novel interventions is also identifiable.*

Corollary 3.6 crucially relies on Assumption 3.5 to achieve separation among all pairs of latents, thanks to Theorem 3.4, hence achieving full identifiability and extrapolation to all unseen interventions.

*Remark* 3.7 (unknown sparse $\boldsymbol{B}$)*.* We note that the identifiability results are primarily driven by the intervention status of $\boldsymbol{Bu}_t$, where the matrix $\boldsymbol{B}$ is unknown. However, a sparse $\boldsymbol{B}$ would ensure that each dimension of the interventional input $\boldsymbol{u}_t$ only affects a few latent dimensions, which has two important consequences. First, an interventional input $\boldsymbol{u}_t$ that is nonzero only in a few dimensions would typically alter only a small subset of latents, enabling identification of intervened latents from unintervened ones based on the

**Theorem 3.4.** Second, designing diverse interventions is usually easier under sparse or known $\boldsymbol{B}$. Therefore, Assumption 3.5 implicitly presupposes the presence of some a priori structure in $\boldsymbol{B}$.

## 4. Results

### 4.1. Identifying motor cortical dynamics in simulations

To illustrate how iSSM leads to identification, we take inspiration from models of the motor cortex. A key observation in the motor cortex is the presence of rotational trajectories (Churchland et al., 2012). From a computational perspective, it has been argued that rotational trajectories provide a basis for motor neuron activations and muscle movements. It has also been argued that the rotational basis provides robustness to noise and interventions (Logiaco et al., 2021). Inspired by these results, multiple dynamical models have been proposed for rotational activities in the motor cortex (Laje & Buonomano, 2013; Sussillo et al., 2015). The first model, called *Rotational Dynamics (RD)* proposes that the motor cortex has underlying rotational dynamics. As a result, in this model, the rotational dynamics are generated within the motor cortex independent of input or feedback activity (Fig. 2I; Sussillo et al. (2015)). Eq. 5 describes the dynamics and emissions of *RD*.

**Rotational Dynamics (RD):**

$$\frac{d\boldsymbol{x}}{dt} = \begin{bmatrix} 0 \\ a\boldsymbol{x}_1 \end{bmatrix} + \boldsymbol{\epsilon}_t, \quad \boldsymbol{y}_t = \begin{bmatrix} \boldsymbol{x}_1\cos(\boldsymbol{x}_2) \\ \boldsymbol{x}_2\sin(\boldsymbol{x}_2) \end{bmatrix} + \boldsymbol{\delta}_t \quad (5)$$

where as before $\boldsymbol{x}, \boldsymbol{y}$ denote the true latents and observations, and $\boldsymbol{\epsilon}, \boldsymbol{\delta}$ represent true latent noise and observation noise. We set the generative noise values to zero in all experiments.

The second model, called *Dynamic Attractor (DA)*, assumes that the underlying dynamics of the motor cortex is a rounded attractor. In this model, the rotational dynamics in motor neurons are generated by some upstream region moving the state along the attractor (Laje & Buonomano, 2013). Eq. 6 describes the dynamics and emissions of *DA*.

**Dynamic Attractor (DA):**

$$\frac{d\boldsymbol{x}}{dt} = \begin{bmatrix} a_1\boldsymbol{x}_1 \\ a_2(1-\boldsymbol{x}_2) \end{bmatrix} + \boldsymbol{\epsilon}_t, \quad \boldsymbol{y}_t = \begin{bmatrix} (1-\boldsymbol{x}_1)\cos(\boldsymbol{x}_2) \\ (1-\boldsymbol{x}_2)\sin(\boldsymbol{x}_2) \end{bmatrix} + \boldsymbol{\delta}_t \quad (6)$$

While these models have distinct characteristics and propose different underlying circuit mechanisms, Galgali et al. (2023) show that the trial averages of these models can be precisely the same, limiting our ability to identify the true dynamics of the motor cortex solely from observational data.

O'Shea et al. (2022) refer to these models as low-dimensional vs. path-following dynamical systems and use

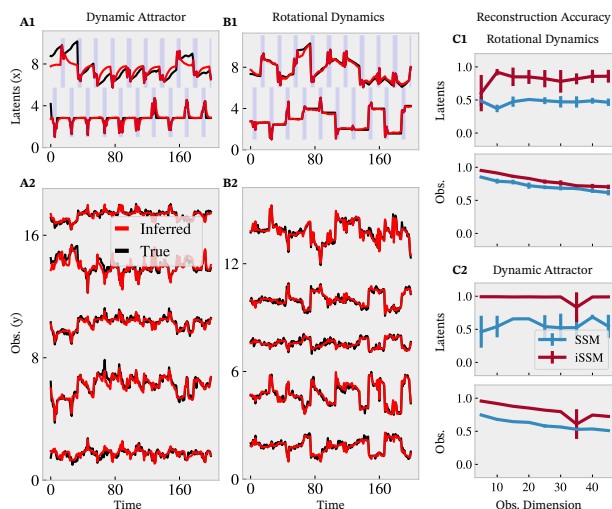

*Figure 3. High-dimensional Models of Motor Dynamics.* To mimic high-dimensional neural responses and low-dimensional latent dynamics, we generated latents from equations 5, 6, and noisy observations from equation 7. (A1,A2) Latents ($\boldsymbol{x}_t$) and observations ($\boldsymbol{y}_t$) for an example trial in 5 dimensions for the *DA* model. (B1,B2) Same as panel A for the *RD* model. (C1) Reconstruction accuracy of the latents (top) and observations (bottom) for the *RD* model using SSM and iSSM models as a function of observation dimension. Results suggest that iSSM not only reconstructs the observations but also recovers the true latents. (C2) Same as C1 for the *DA* model.

an interventional strategy to discover whether the dynamics in the motor cortex follow either of these regimes. Similarly, here we ask if interventional data can distinguish between these models. We generate data from *RD* and *DA* to address this. The latent states $\boldsymbol{x}(t)$ in both models follow linear dynamics, while the observation model in both cases is highly nonlinear. Therefore, recovering the true latents is not a trivial task.

During data generation, we apply repeated interventions interleaved by resting periods for the network to return to its stationary state. The dynamics of latents and observations are shown in Fig. 2A,D. Although in the absence of interventions both models produce the same trajectories, one can observe that the interventional trajectories exhibit distinct characteristics (Fig. 2A).

Consistent with O'Shea et al. (2022), our results suggest that in the presence of interventional data using the iSSM model, one can identify the underlying dynamics and emissions (Fig. 2B,C) and recover the true latent variables (Fig. 2D). This recovery continues to improve as we collect more interventional data, emphasizing the importance of perturbation experiments in causal hypothesis testing (Fig. 2E). The reproducibility details are included in the Appendix B.

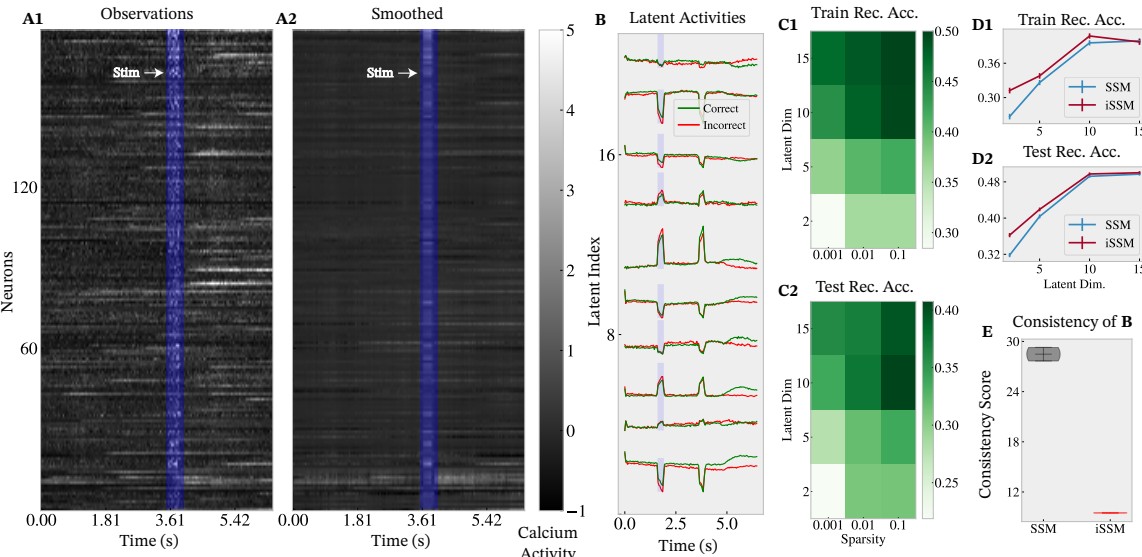

*Figure 4. Results on Mice Dataset.* (A1) Calcium responses of ALM neurons in an example trial during stimulation (specified by the shaded blue band and the white arrow). (A2) Smoothed responses given by the mean posterior of the model. (B) Mean latents discovered by iSSM for correct (green) vs. incorrect (red) trials. The dynamics of the latents distinguish between correct and incorrect trials without any prior knowledge of the behavior. (C1,C2) Training (C1) and test (C2) reconstruction accuracy of iSSM for a range of hyperparameters (varying number of latents and the sparsity of $\boldsymbol{B}$). Both higher sparsity and a larger number of latents improve the accuracy. (D1,D2) Comparison between SSM (blue) and iSSM (red) for the optimal selection of the hyperparameters for training (D1) and test (D2) data. (E) Consistency score of matrix $\boldsymbol{B}$ across random initializations of the model parameters for SSM and iSSM, showing that iSSM leads to better identifiability (see section 4.5).

## 4.2. Simulations of high-dimensional motor activity

To mimic high-dimensional neural responses in the motor cortex, we projected data from the *DA* and *RD* models to higher dimensions. For each run, we generated a random orthogonal matrix $\boldsymbol{C} \in \mathbb{R}^{D \times N}$ and used it in the generation of $\boldsymbol{y}_t$ in the following way:

**High-dimensional Simulations:**

$$\boldsymbol{y}_t = \boldsymbol{C} \begin{bmatrix} \boldsymbol{x}_1 \cos(\boldsymbol{x}_2) \\ \boldsymbol{x}_2 \sin(\boldsymbol{x}_2) \end{bmatrix} + \boldsymbol{\delta}_t, \quad \boldsymbol{y}_t = \boldsymbol{C} \begin{bmatrix} (1 - \boldsymbol{x}_1) \cos(\boldsymbol{x}_2) \\ (1 - \boldsymbol{x}_2) \sin(\boldsymbol{x}_2) \end{bmatrix} + \boldsymbol{\delta}_t$$
$$(7)$$

for *RD* and *DA* respectively. In this case, $\boldsymbol{\delta}_t \in \mathbb{R}^N$ is a high-dimensional noise added to the responses. We set the standard deviation of noise to $0.1$ for all high-dimensional experiments.

An example of the latents $\boldsymbol{x}_t$ and 5-dimensional responses $\boldsymbol{y}_t$ is shown in Fig. 3A,B. We generated data for a varying number of observation dimensions and applied SSM and iSSM models (Fig. 3C). While both SSM and iSSM achieve high reconstruction accuracy, only iSSM is able to identify the true latents (Fig. 3C1,C2). The error bars represent the standard deviation of the reconstruction accuracy across 3 seeds of 20 randomly generated trials for each observation dimension. Other generative and inference parameters are the same as the previous section (see Appendix B).

## 4.3. Identifying dynamics underlying short-term memory in mice

Persistent activity is a hallmark of short-term memory across species (Romo et al., 1999; Fuster & Alexander, 1971). How can a network of neurons produce activities in response to an input stimulus that is maintained after the stimulus is removed? Multiple network mechanisms are proposed to underlie persistent activity. Among those, one popular model is known as *Functionally Feedforward (FF)* model (Goldman, 2009). *FF* assumes that the network consists of a few smaller subnetworks that are connected in a feedforward manner. Since these subnetworks do not necessarily need to form a spatial cluster in the brain, experimentally finding footprints of this type of connectivity is not feasible. However, the theoretical properties of the model have been well-studied. For example, it is commonly argued that *FF* results in robustness to structural noise (Qian et al., 2024). An alternative model for the persistent activity is known as *Line Attractor (LA)* model (Seung, 1996). Under *LA* circuit model, the activity of an upstream region pushes the state of the circuit along the line attractor, and the dynamics preserve the state until a new input has arrived (see Appendix A.1 for results on models of working memory).

We applied iSSM to a public dataset of targeted photostimulation in the anterior lateral motor cortex (ALM) of mice

during a short-term memory task (Daie et al., 2021). The task included a sample epoch where an auditory cue guided the mice for a left vs. right cue to get a water reward. The sample epoch was followed by a delay epoch of 3 seconds, where the mice needed to engage working memory to keep track of the guided cue. Finally, during the response period, the mice received the reward if the lick direction was correct. The photostimulation was delivered during the delay period for a short amount of time, started simultaneously with the delay period, or after 1 or 2 seconds.

Calcium recordings were done in 179 identified neurons for 77 repeated trials (Fig. 4A). There were 8 photostimulation channels targeted to stimulate neurons according to their response selectivity.

We set the dimension of interventional inputs $u_t$ to the number of photostimulation channels and fitted the stimulation matrix $B$ with a sparsity penalty. The smoothed and denoised neural activities are shown in Fig. 4A2. The reconstruction accuracy of the data for both training and testing trials for iSSM was larger than the baseline SSM model across a range of latent dimensions (Fig. 4D). Furthermore, the latents learned by the model show distinct mean trajectories for correct vs. incorrect trials, suggesting that they capture behaviorally meaningful dynamics (Fig. 4B). The optimal latent dimension and sparsity penalty for this dataset were found based on cross-validation (Fig. 4C).

### 4.4. Generalizing to test interventions in primates

Understanding network dynamics to control behavior has been a longstanding challenge in neuroscience. The overarching goal is to deliver targeted stimulation to a network of neurons to steer the dynamics or the behavior towards a pre-determined outcome (Haimerl et al., 2023; Jou et al., 2023). A first step towards understanding the circuit effects or behavioral influences of network manipulations is to build models that can predict the response to interventions. The space of possible interventions is combinatorial and intractable to cover. Therefore, an alternative approach is to build models that can generalize to unseen interventions.

We showed theoretically in section 3.3 that iSSM has this property. Concretely, if we fit the iSSM model to interventional data, where the dataset consists of a small set of canonical interventions, the model can generalize to unobserved interventions. To validate this empirically, we showed results on a synthetic dataset (Fig. 2). Here we want to test whether these results hold in a real biological dataset.

The dataset consisted of electrophysiological recordings using electrode arrays implanted on the prefrontal cortex of macaque monkeys during quiet wakefulness (resting) while the animals were sitting awake in the dark. The electrode array included 96 electrodes that were also used for de-

livering micro-circuit electrical stimulations (Nejatbakhsh et al., 2023). We analyzed 6 datasets, 3 with only observational data and 3 with a combination of observational and interventional data.

In Fig. 5A, we show firing rates recorded from each of the 96 electrodes for an interventional session. In each interventional session, two electrodes were repeatedly stimulated while recordings were performed from all other electrodes. We apply iSSM to this dataset and infer the latents and parameters. Fig. 5A2 shows a sample from the inferred model closely matching the data. The reconstruction accuracy on the training and testing sessions is larger for iSSM compared to baseline SSM across a range of latent dimensions (Fig. 5E), suggesting that the model can better generalize to test trials. To find the optimal hyperparameters, we performed cross-validation. The results are shown in Fig. 5D.

### 4.5. Performance Metrics

Here, we briefly describe the metrics used throughout the experiments in the paper.

**Reconstruction accuracy:** The reconstruction accuracy is defined as the correlation coefficient between the true and reconstructed signals (latents or observations). For latents, it can only be computed for synthetic experiments where ground truth exists. This metric is commonly used in the causal representation learning literature for assessing the identifiability of the latents (Khemakhem et al., 2020a;b; Song et al., 2024a).

**Consistency score:** A hallmark of identification is robustness to initialization. To test whether the iSSM results in identifiability, we ran the model several times with different random initializations. We compute the consistency score across random initialization by first aligning the columns of $B$ across runs to account for permutation invariance of the latents, followed by computing the Euclidean distance between aligned matrices. The aligned distances are considerably smaller for iSSM compared to SSM, providing evidence for the identifiability (e.g., Fig. 4E).

## 5. Discussion

**Summary** Here we proposed iSSM, a framework for joint modeling of observational and interventional data. We provided theoretical results showing that the iSSM model, when fitted on interventional data, leads to the identifiability of latents as well as dynamics and emissions. To illustrate iSSM's applicability, we showed results on 3 different examples covering a range of assumptions. The first example was a synthetic dataset with linear dynamics and nonlinear emissions. The second example was calcium recordings from the mouse ALM region with targeted photostimulation

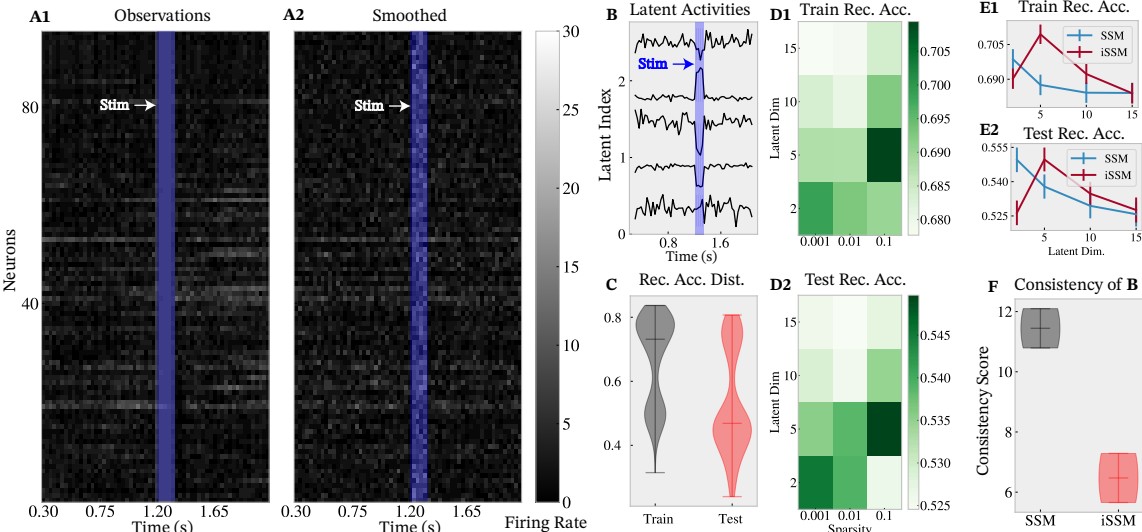

*Figure 5. Results on Primate Dataset.* (A1) Unit firing rates for a training interventional session (stimulation times are shown by the blue band and white arrow). (A2) Sample responses for the same trial inferred by iSSM. (B) iSSM's latent dynamics show differential responses during stimulation vs. resting periods. (C) The distribution of training and test reconstruction accuracy across trials. (D1,D2) Training (D1) and test (D2) mean reconstruction accuracy is shown with varying hyperparameters (number of latents and sparsity parameter of $\boldsymbol{B}$). (E1,E2) Comparison between SSM (blue) and iSSM (red) for an increasing number of latents. Reconstruction accuracies are shown for training (E1) and test (E2) trials. Both SSM and iSSM benefit from a larger number of latents, with iSSM consistently outperforming SSM. (F) Consistency of the inferred $\boldsymbol{B}$ across random initializations showing better identifiability for iSSM (see section 4.5).

delivered by channels that targeted groups of neurons. The third example was electrophysiological recordings from the macaque monkey's prefrontal cortex with micro-stimulation delivered by the same recording electrodes. In all cases, our results show impressive generalization capabilities and parameter recovery, confirming our theoretical results.

**Limitations** We identify several limitations of our work. First, in this work, we focused on a generative model that has linear dynamics. While the inference model can still capture nonlinearities through its recognition network, explicitly modeling nonlinearities and providing theoretical results is an important limitation of this work.

Second, as depicted in Fig. 1, we think of interventions as forcing the neural state out of its low-dimensional manifold. Therefore, it is a valid question how can we still assume the low dimensionality of dynamics. Our argument is the following: (1) While the observational data is low-dimensional, we think that interventional data is higher-dimensional but still lives in a much lower dimension than the number of neurons. This is simply because certain configurations of neural states are not biologically possible. (2) The optimal latent dimension depends on the sparsity of the $\boldsymbol{B}$ as well as the amount of data available. With larger datasets and more interventions, we conjecture that increasing the number of latents always improves the results.

Third, a critical modeling assumption that we made is that interventions directly affect the latents. Furthermore, our theoretical results require performing interventions on individual latents through neurons that are causally linked to them. Further experimental validation is required to test whether this assumption is valid in neural data.

Fourth, here we rely on variational inference (VI). While VI is commonly used in state space modeling, it is an approximate method. Furthermore, the choice of the recognition network (in our case, LSTMs) can amplify the approximation gap. Better inference algorithms and more powerful architectures can further improve the identifiability results.

Finally, this work establishes a framework for modeling neural data under interventions and is intended to motivate future investigations into its limitations and extensions.

## Acknowledgments

YW was supported by the Office of Naval Research under grant N00014-23-1-2590, the National Science Foundation under grant No. 2231174, No. 2310831, No. 2428059, No. 2435696, No. 2440954, and a Michigan Institute for Data Science Propelling Original Data Science (PODS) grant. The authors also acknowledge helpful discussions with Kayvon Daie, Roozbeh Kiani, Alex H Williams, Timothy Kim, and constructive feedback from the reviewers.

## Impact Statement

The modeling presented in this work aims to offer a methodological framework for advancing our understanding of neural computation under interventions. While the analysis of electrophysiological and calcium imaging data holds potential for long-term impact on the understanding of neurological disorders across species, such implications lie beyond the scope of this theoretical modeling. We therefore anticipate no immediate societal consequences from this study.

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

# A. Experimental Results

## A.1. Models of Working Memory

Various sources of non-identifiability can make it challenging to recover the true latents and dynamics in neural data. We elaborate on two of these sources here. First, neural recordings are undersampled, meaning that from a large pool of neurons involved in the computation only a small fraction are recorded. Undersampling or partial observation is indeed a significant origin of non-identifiability as discussed in the literature (Beiran & Litwin-Kumar, 2024). Second, non-identifiability can be caused by the mixing of input-driven and recurrent activity in the network (Galgali et al., 2023). Lipshutz et al. show that noise correlations can be used to disentangle input-driven and recurrent activity. Here, we take a complementary approach and investigate whether interventions can help improve identifiability. A recent theoretical study investigates the effect of partial observations in the context of persistent activity (Qian et al., 2024).

Qian et al. (2024) presented two alternative models of working memory, functionally feedforward (*FF*) and low-rank connectivity (*LR*). *FF* is characterized by the existence of sub-circuits in the network that sequentially process their inputs and pass the outputs to the next sub-circuits in a feedforward manner (Fig. 6A1). On the other hand, *LR* models perform their computations recurrently by utilizing line-attractor type dynamics (Fig. 6A2). Importantly, this study shows that when the system is partially observed, observational models (e.g. SSM) cannot distinguish between the two models. Specifically, they show that linear dynamical system (LDS) models have a built-in bias for characterizing the dynamics as *FF*, regardless of the underlying dynamics (Qian et al., 2024).

Here we ask whether interventional models (e.g. iSSM) can help identify the dynamics. To address this, we generated interventional data from *FF* and *LR* models with 5 neurons (shown in Fig. 6B,C). The interventions are performed on individual neurons interleaved by a resting duration (blue shades in Fig. 6B). We then used iSSM with $N = D = 5$ and set $B$ to identify to check if we can recover the underlying dynamics. Setting the matrix $B$ to identity amounts to assigning one latent per neuron, an important variation of iSSM that enables modeling interventions on individual neurons. In Fig. 6D-F we show that while both SSM and iSSM successfully reconstruct the observational data (Fig. 6D), only iSSM is able to identify the dynamics matrices (Fig. 6E,F). These results suggest that the ability to perform interventions on individual neurons accompanied by models that leverage the interventional data enable uncovering more details of the causal flow of information in a circuit of neurons.

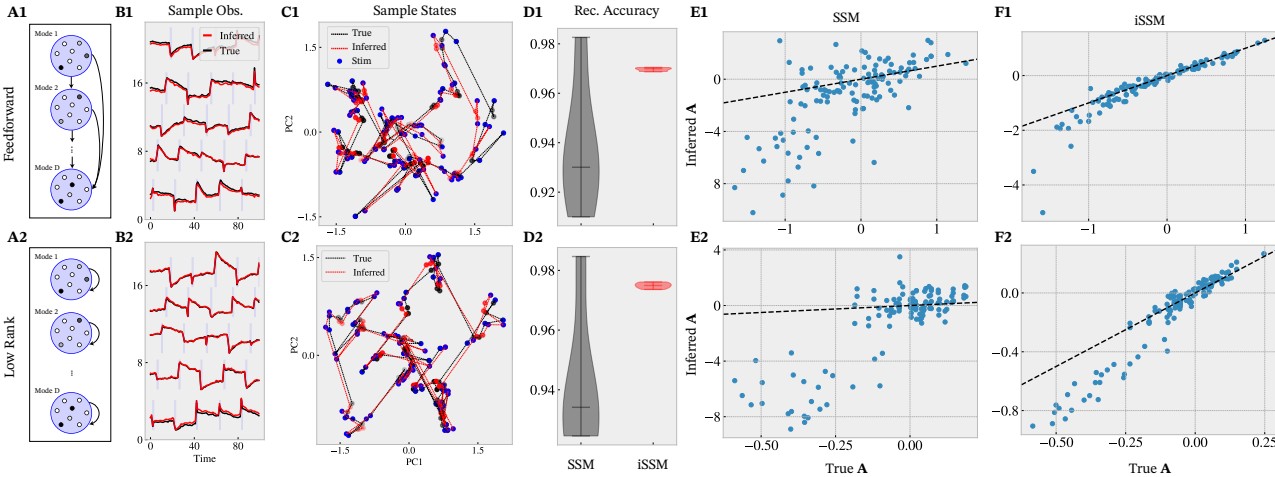

*Figure 6. Models of Working Memory.* Following (Qian et al., 2024) here we generate data from feedforward (*FF*) and low-rank (*LR*) models of working memory to test whether iSSM can recover the true underlying flow field parameterized by matrix $A$. (A1,A2) Schematics of circuit mechanisms for *FF*, *LR*. (B1,B2) True (black) and inferred (red) interventional signals generated from *FF* (B1) and *LR* (B2) in 5 dimensions. The inference is done using iSSM with linear dynamics and linear observations. Blue shades represent interventions. (C1,C2) Same data shown in the top 2 PC space with blue dots representing interventions. (D1,D2) Comparison of observational reconstruction accuracy using SSM vs. iSSM. While both SSM and iSSM achieve a high accuracy for both *FF* and *LR* systems, iSSM estimator has lower variance and is more accurate on average. (E1,E2,F1,F2) Scatter plot of the true values of $A_{ij}$ (x-axis) vs. their inferred values (y-axis) for *FF* (E1,F1) and *LR* (E2,F2) models using SSM (E1,E2) and iSSM (F1,F2) across 3 runs. This result suggests that iSSM is able to leverage interventional data to identify the matrix $A$.

## B. Reproducibility Details

### B.1. Generative Parameters for Synthetic Experiments

In table B.1 we include the parameters used for generating synthetic datasets for Fig. 2. The stimulation signals $u_t$ are parameterized by the following parameters: *Stim Amp* the overall amplitude of the interventional inputs; *Stim Noise*: the standard deviation of the zero-mean gaussian noise added to the interventional inputs to satisfy the assumption3.1; *Stim to Rest Ratio* the ratio of the amount ot time spend on delivering stimulation signals vs. resting; *Stim Rep*: the number of stimulation repetitions.

We generate data for a duration of $T$ seconds, for $K$ trials, using the time increment $dt$. The number of observation dimensions and latent dimensions are denoted by $N, D$ respectively. The equations governing the generative process of the data given the interventional inputs $u_t$ are included in Eq. 3 with parameters $a$ for the Rotational Dynamics and $a_1, a_2$ for the Dynamic Attractor.

Table 1.  *Generative parameters for synthetic experiments.*

| | T | K | N | D | dt | Params | Model Noise | Initialization | Stim Amp | Stim Noise | Stim to Rest Ratio | Stim Rep |
|---|---|---|---|---|---|---|---|---|---|---|---|---|
| **Rotational Dynamics** | 20 | 10 | 2 | 2 | 0.05 | $a = 1.2$ | 0 | $\mathcal{N}(0,1)$ | 1 | 0.5 | 1 | 10 |
| **Dynamic Attractor** | 20 | 10 | 2 | 2 | 0.05 | $a1 = -20$ $a2 = 1.2$ | 0 | $\mathcal{N}(0,1)$ | 1 | 0.5 | 1 | 10 |

### B.2. Initialization and Inference Parameters

Table B.2 contains the parameters for initializing our generative process as well as inference and optimization parameters. The *Emission* model is a fully connected (*FC*) neural network with $H = 100$ hidden units. For Poisson observation model, we include an additional `softplus` transformation to map the emission outputs to positive values. The *Obs.* column in the table determines what observation noise model was used in each experiment. For the synthetic experiments, we set the $B$ matrix to identity without fitting it, therefore we did not use any *Sparsity* regularization. For real data experiments we used cross-validation to find the optimal value. For the primate dataset, the training data is the first half of each session and the test data is the second half. For the mice dataset the training data is the lick right trials while the test data is lick left trials. The optimization parameters include the learning rate denoted by *Optim LR*, number of iterations denoted by *Optim (Iter)*, and the number of hidden units for the LSTM that parameterizes the mean and variance of the variational posterior denoted by *LSTM (H)*. Furthermore, we include the initialization values in table B.2. The initialization parameters include the noise covarinace of the dynamics' initial step denoted by *Init ($x_0$ noise)*; the covariance of the LDS denoted by *Init (LDS $\sigma$)*; the matrices $A, B$; and the covariance scaling of the likelihood model referred to as *Init (LL $\sigma$)*.

Table 2.  *Initialization and inference parameters for synthetic and real data experiments.*

| | Emission | dt | Obs. | Sparsity ($s$) | Optim (LR) | Optim (Iter) | LSTM ($H$) | Init ($x_0$ noise) | Init (LDS $\sigma$) | Init ($A$, $B$) | Init (LL $\sigma$) |
|---|---|---|---|---|---|---|---|---|---|---|---|
| **Rotational Dynamics** | FC (H=100) | 0.05 | Normal | N/A | 0.01 | 1000 | 10 | 0.05 | 0.05 | $\mathcal{N}(0,1)$ $I$ | 0.05 |
| **Dynamic Attractor** | FC (H=100) | 0.05 | Normal | N/A | 0.01 | 1000 | 10 | 0.05 | 0.05 | $\mathcal{N}(0,1)$ $I$ | 0.05 |
| **Mice Dataset** | FC (H=100) | 0.05 | Normal | X-Val | 0.001 | 1000 | 10 | 0.05 | 0.05 | $\mathcal{N}(0,1)$ $\mathcal{N}(0,1)$ | 0.05 |
| **Primate Dataset** | FC (H=100) `softplus` | 1 | Poisson | X-Val | 0.01 | 1000 | 10 | 0.05 | 0.05 | $\mathcal{N}(0,1)$ $\mathcal{N}(0,1)$ | N/A |

## C. Proof of Theorem 3.4

We consider the interventional state space model (iSSM),

$$\boldsymbol{y}_t \sim P(\boldsymbol{y}_t | f_{\boldsymbol{\theta}}(\boldsymbol{x}_t)), \tag{8}$$

$$\boldsymbol{x}_{t+1} = 1\{\boldsymbol{B}\boldsymbol{u}_t = 0\} \otimes \boldsymbol{A}\boldsymbol{x}_t + \boldsymbol{B}\boldsymbol{u}_t + \boldsymbol{\epsilon}_t. \tag{9}$$

**Step I: Identifying the distribution of $z_t \triangleq f_{\boldsymbol{\theta}}(\boldsymbol{x}_t)$.**   We begin with identifying the marginal distribution of $P(\boldsymbol{z}_t)$ from $P(\boldsymbol{y}_t)$. The core assumption we rely on in this step is bounded completeness, which we define in Assumption 3.1

The bounded completeness of $P(\boldsymbol{y}_t | \boldsymbol{z}_t)$ implies that $P(\boldsymbol{z}_t)$ is identifiable from $P(\boldsymbol{y}_t)$. It is because $P(\boldsymbol{z}_t)$ must be the unique solution to the integral equation $\int P(\boldsymbol{y}_t | \boldsymbol{z}_t) P(\boldsymbol{z}_t) \mathrm{d}\boldsymbol{z}_t = P(\boldsymbol{y}_t)$. Specifically, if there are two solutions to this equation $\hat{P}_1(\boldsymbol{z}_t), \hat{P}_2(\boldsymbol{z}_t)$, then they must be equal. It is due to the bounded completeness of $P(\boldsymbol{y}_t | \boldsymbol{z}_t)$: the two solutions must satisfy $int P(\boldsymbol{y}_t | \boldsymbol{z}_t)[\hat{P}_1(\boldsymbol{z}_t) - \hat{P}_2(\boldsymbol{z}_t)] \mathrm{d}z_t = 0$, which implies $\hat{P}_1(\boldsymbol{z}_t) = \hat{P}_2(\boldsymbol{z}_t)$.

**Step 2: Affine identification of $f_{\boldsymbol{\theta}}(\cdot)$ and $P(\{\hat{\boldsymbol{x}}_t\}_{t \in T})$.**   In this step, we establish the affine identification of the mixing function $f_{\boldsymbol{\theta}}(\cdot)$ by invoking Theorem 3.5 of Balsells-Rodas et al. (2023): identifying $f_{\boldsymbol{\theta}}(\cdot)$ from $P(f_{\boldsymbol{\theta}}(\boldsymbol{x}_t))$ is a special case of identifying the mixing function in a switching dynamical system.

To enable identification, we require Assumption 3.2. In particular, the mixing function should be a piece-wise linear function.

**Lemma C.1** (Theorem 3.5 of Balsells-Rodas et al. (2023)). *Under Assumption 3.2, the mixing function $f_{\boldsymbol{\theta}}(\cdot)$ and the latent distribution $P(\{\hat{\boldsymbol{x}}_t\}_{t \in T})$ can be identified from $P(f_\theta(\boldsymbol{x}_t))$ up to affine transformation.*

This lemma is an instantiation of Theorem 3.5 in Balsells-Rodas et al. (2023) in the special case of linear transition dynamics.

**Step 3: Identification of $\boldsymbol{x}_t$ via interventions.**   The previous step shows that we can identify $\boldsymbol{x}_t$ up to affine transformation. In this step, we show that, if two solutions of $\boldsymbol{x}_t$ are affine transformations of each other, they must coincide up to block affine transformation, if they agree on the interventional distributions, under Assumption 3.3. This argument implies that the interventional distributions can identify $\boldsymbol{x}_t$ (up to block-wise permutation, and shifting and scaling.)

Concretely, consider two sets of latent variables $\{\boldsymbol{x}_t\}_{t \in T}$ and $\{\hat{\boldsymbol{x}}_t\}_{t \in T}$ where they are affine transformations of each other

$$\hat{\boldsymbol{x}}_t = M\boldsymbol{x}_t + c, \forall t. \tag{10}$$

Suppose both sets satisfy Equation (23) across all intervention environments, namely,

$$\boldsymbol{x}_{t+1} = 1\{\boldsymbol{B}\boldsymbol{u}_t = 0\} \otimes \boldsymbol{A}\boldsymbol{x}_t + \boldsymbol{B}\boldsymbol{u}_t + \boldsymbol{\epsilon}_t, \tag{11}$$

$$\hat{\boldsymbol{x}}_{t+1} = 1\{\hat{\boldsymbol{B}}\boldsymbol{u}_t = 0\} \otimes \hat{\boldsymbol{A}}\hat{\boldsymbol{x}}_t + \hat{\boldsymbol{B}}\boldsymbol{u}_t + \hat{\boldsymbol{\epsilon}}_t, \tag{12}$$

where both $\boldsymbol{\epsilon}_t, \hat{\boldsymbol{\epsilon}}_t$ are i.i.d over time. Then we will prove that $M = \Lambda\Pi$, where $\Lambda$ is an invertible block-diagonal matrix (with a $\mathbf{1}^\top 1\{\boldsymbol{B}\boldsymbol{u}_t = 0\}$-sized block and a $\mathbf{1}^\top 1\{\boldsymbol{B}\boldsymbol{u}_t \neq 0\}$-sized block), and $\Pi$ is a permutation matrix.

**Single intervention case.**   Below we first consider the simplest case where only one latent variable is intervened, i.e. $\mathbf{1}^\top 1\{\boldsymbol{B}\boldsymbol{u}_t = 0\} = 0$. We will entend to general interventions later.

We achieve identification using the following observation. Suppose the $j$th latent $x_{t,j}$ was intervened in an environment, namely $1\{(\boldsymbol{B}\boldsymbol{u}_t)_j = 0\} = 0$. Then we have

$$x_{t+1,j} = (\boldsymbol{B}\boldsymbol{u}_t)_j + \epsilon_{t,j} \qquad \text{for } t \in \{t : \mathbf{B}\mathbf{u_t} \neq 0\}, \tag{13}$$

and thus $x_{t+1,j} \perp \boldsymbol{x}_t$ for all $t$. The reason is that the intervention set $\mathbf{x}_{t+1,j}$ to be $(\boldsymbol{B}\boldsymbol{u}_t)_j$ plus a random noise component, hence independent of all components of $\mathbf{x}_t$.

Below we argue that, if we also find a component $j'$ of $\hat{\boldsymbol{x}}_{t+1}$ such that $\hat{x}_{t+1,j'} \perp \hat{\boldsymbol{x}}_t$, then $M_{j',-j} = 0$, i.e. $\hat{x}_{t+1,j'}$ must be an affine transformation of $x_{t+1,j}$.

To make this argument, we write

$$\hat{\boldsymbol{x}}_{t+1,j'} = M_{j',-j}^\top \boldsymbol{x}_{t+1,-j} + M_{j',j} x_{t+1,j} + c_{j'}, \tag{14}$$

$$\hat{\boldsymbol{x}}_{t,j'} = M_{j',-j}^\top \boldsymbol{x}_{t,-j} + M_{j',j} x_{t,j} + c_{j'}. \tag{15}$$

Then since $\hat{x}_{t+1,j'} \perp \hat{\boldsymbol{x}}_t$, we have that

$$Cov(\hat{x}_{t+1,j'}, \hat{\boldsymbol{x}}_t) = 0. \tag{16}$$

This implies

$$0 = Cov(\hat{x}_{t+1,j'}, \hat{\boldsymbol{x}}_t) \tag{17}$$

$$= Cov(M_{j',-j}^\top \boldsymbol{x}_{t+1,-j} + M_{j',j} x_{t+1,j}, M_{j',-j}^\top \boldsymbol{x}_{t,-j} + M_{j',j} x_{t,j}) \tag{18}$$

$$= Cov(M_{j',-j}^\top \boldsymbol{x}_{t+1,-j}, M_{j',-j}^\top \boldsymbol{x}_{t,-j}) + Cov(M_{j',-j}^\top \boldsymbol{x}_{t+1,-j}, M_{j',j} x_{t,j})$$

$$+ Cov(M_{j',j} x_{t+1,j}, M_{j',-j}^\top \boldsymbol{x}_{t,-j}) + Cov(M_{j',j} x_{t+1,j}, M_{j',j} x_{t,j}) \tag{19}$$

$$= Cov(M_{j',-j}^\top \boldsymbol{x}_{t+1,-j}, M_{j',-j}^\top \boldsymbol{x}_{t,-j}) + Cov(M_{j',-j}^\top \boldsymbol{x}_{t+1,-j}, M_{j',j} x_{t,j}). \tag{20}$$

The last equation is due to Equation (13). It implies that $M_{j',-j} = 0$. The reason is that Assumption 3.3 guarantees one of $V, W$ must be zero, if we write Equation (20) as $Cov(V^\top \boldsymbol{x}_{t+1}, W^\top \boldsymbol{x}_t) = 0$, where $W = M_{j'}$ and $V$ is a matrix where $V_j = 0$ and $V_{-j} = M_{j',-j}$, which implies $M_{j',-j} = 0$. Thus we have that the $j'$th dimension of $\hat{\boldsymbol{x}}_t$ that achieves the independence property is mapped to the $j$th dimension of $\boldsymbol{x}_t$ up to scaling and shifting; one can separate out the intervened latent from the uninterven ed ones up to permutation, and coordinate-wise shifting and scaling.

We note that the use of covariance is not essential to the proof. Rather, any statistic that can characterize independence relationships (e.g. moment generating function) apply here and can be leveraged to make this identification argument.

**Multiple intervention case.** We next extend to the case when multiple latents $j \in \mathcal{I}$ were intervened, which gives $x_{t+1,\mathcal{I}} \perp \boldsymbol{x}_t$ in the true latent space.

We find a latent space (i.e. $\hat{x}_t$) for the iSSM model that is compatible with the interventional data while maximizing the cardinality of $\mathcal{I}'$ that satisfies a similar independence relationship as the true intervened latents $\hat{x}_{t+1,\mathcal{I}'} \perp \hat{\boldsymbol{x}}_t$. In other words, among all candidate $\hat{x}_t$ that can fit the interventional data, we find the one that has the most number of dimensions independent of $\hat{\boldsymbol{x}}_t$. The faithfulness assumption (Assumption 3.3) implies that one cannot use linear transformations to induce additional independence relatinoship, hence $|\mathcal{I}'| = |\mathcal{I}|$.

We then leverage the linear identifiability from step 2 that gives

$$\{\hat{\Pi} \hat{\boldsymbol{x}}_{t+1}\}_{\mathcal{I}} = M_{\mathcal{I},-\mathcal{I}} \hat{\boldsymbol{x}}_{t+1,-\mathcal{I}} + M_{\mathcal{I},\mathcal{I}} x_{t+1,\mathcal{I}} + c_{\mathcal{I}}. \tag{21}$$

This step mimics Equation (14) except we use an additional permutation matrix $\hat{\Pi}$ to align the $\mathcal{I}'$ entries in $\hat{\boldsymbol{x}}$ with the $\mathcal{I}$ entries in $\boldsymbol{x}$. This alignment is for notational convenience.

Then following the same argument as in Equation (17), $Cov(\{\hat{\Pi} \hat{\boldsymbol{x}}_{t+1}\}_{\mathcal{I}}, \hat{\boldsymbol{x}}_t) = 0$ implies that $M_{\mathcal{I},-\mathcal{I}} = 0$. That is, we can successfully separate out the intervened latents from the un-intervened latents: the intervened latents $\mathcal{I}$ were mapped to $\mathcal{I}'$ in the estimated latent space up to permutation, and block-wise shifting and scaling.

As a consequence of identifying the parameters of the iSSM up to blockwise affine transformation, we can predict the observation distributions for novel unseen interventions $\boldsymbol{u}_t$ as long as the interventions touch upon only those latent nodes that are in different blocks; these latents are already separated out: $\{\Lambda\Pi\}_{ij} = 0$ for any $i, j \in \{k : \{\boldsymbol{B}\boldsymbol{u}_t^{\text{new}}\}_k \neq 0\}$. In other words, the submatrix restricted to the intervened latents in the new intervention $\{\Lambda\Pi\}_{\mathcal{I}^{\text{new}} \times \mathcal{I}^{\text{new}}}$ must be diagonal, ensuring that interventions performed on these latents are properly modeled and predicted.

**Sufficiently diverse interventions.** We finally extend Theorem 3.4 to a set of sufficiently diverse interventions. The unordered pairs condition (Assumption 3.5) ensures that $M_{ij} = 0$ for any pair of $i, j$ latents, following the same argument as in Equation (17), since we can separate out any pair of latents as long as they are not always either being intervened at the same time or being un-intervened at the same time. Hence, one can identify the latents up to permutation, and coordinate-wise shifting and scaling, i.e. $\hat{\boldsymbol{A}} = \boldsymbol{A}\Lambda^*\Pi + \boldsymbol{c}$ for some diagonal $\Lambda^*$. It further implies that one can identify the latent dynamics matrix $\boldsymbol{A}$ also up to permutation, and coordinate-wise shifting and scaling. It also enables us to predict for all unseen interventions given the full identifiability of the dynamics,

$$\boldsymbol{y}_t \sim P(\boldsymbol{y}_t | f_{\boldsymbol{\theta}}(\boldsymbol{x}_t)), \tag{22}$$

$$\boldsymbol{x}_{t+1} = \mathbb{1}\{\boldsymbol{B}\boldsymbol{u}_t = 0\} \otimes \boldsymbol{A}\boldsymbol{x}_t + \boldsymbol{B}\boldsymbol{u}_t + \boldsymbol{\epsilon}_t. \tag{23}$$

## C.1. Inference Details

Our statistical model is described by the following probabilistic decomposition.

$$p_{\boldsymbol{\Theta}}(\boldsymbol{x}_{1:T}, \boldsymbol{y}_{1:T}, \boldsymbol{u}_{1:T}) = p(\boldsymbol{B})p_{\boldsymbol{\Theta}}(\boldsymbol{x}_0) \prod_{t=1}^{T} p_{\boldsymbol{\Theta}}(\boldsymbol{x}_{t+1}|\boldsymbol{x}_t, \boldsymbol{u}_t)p_{\boldsymbol{\Theta}}(\boldsymbol{y}_t|\boldsymbol{x}_t)$$

where $\boldsymbol{\Theta} = \{\boldsymbol{A}, \boldsymbol{B}, \boldsymbol{Q}, \boldsymbol{\theta}, \boldsymbol{R}, \boldsymbol{\mu}_0, \boldsymbol{Q}_0\}$ contains all generative parameters of the model and $\boldsymbol{\mu}_0, \boldsymbol{Q}_0$ are the mean and covariance of the initial condition distribution $p_{\boldsymbol{\Theta}}(\boldsymbol{x}_0)$. We may include an independent Laplace prior over the elements of $\boldsymbol{B}$ to encourage its sparsity.

Notice that some of these parameters depend on the specific instantiation of the model. For example, for normal observations $\boldsymbol{R} \in \mathbb{R}^{N \times N}$ is the observation covariance whereas the observation mean is given by the nonlinear transformation $f_{\boldsymbol{\theta}}(\boldsymbol{x})$. For the model with Poisson observations there are no observational parameters since the rate of the Poisson distribution is generated by applying a positive transform (e.g. `softplus`) to the output of the nonlinear transform $f_{\boldsymbol{\theta}}(\boldsymbol{x})$. In addition, for some instantiations of the model we set $\boldsymbol{B}$ to a predefined matrix and do not optimize it. For example, when we want to explicitly model the causal interactions between neurons, we set $D = N$ and set $\boldsymbol{B}$ to the identity matrix.

Given a dataset of $K$ trials of length $T$ denoted by $\left\{\boldsymbol{y}_{1:T}^{(k)}, \boldsymbol{u}_{1:T}^{(k)}\right\}_{k=1}^{K}$ our goal is to approximate the posterior distribution $p_{\boldsymbol{\Theta}}(\boldsymbol{x}_{1:T}|\boldsymbol{y}_{1:T}, \boldsymbol{u}_{1:T})$ while finding the optimal generative parameters that fit the data the best denoted by $\hat{\boldsymbol{\Theta}}$. Since marginal likelihood optimization and analytical posterior inference in this model is not tractable, we follow variational inference (VI), where the goal is to find a distribution $q_{\boldsymbol{\Phi}}(\boldsymbol{x}_{1:T}|\boldsymbol{y}_{1:T}, \boldsymbol{u}_{1:T})$ that best approximates the posterior distribution while optimizing the parameters $\boldsymbol{\Theta}$ via maximizing the evidence lower bound (ELBO).

$$\mathcal{L}(\boldsymbol{\Phi}, \boldsymbol{\Theta}) = \mathbb{E}_q\left[\log p_{\boldsymbol{\Theta}}(\boldsymbol{x}_{1:T}, \boldsymbol{y}_{1:T}, \boldsymbol{u}_{1:T}) - \log q_{\boldsymbol{\Phi}}(\boldsymbol{x}_{1:T}|\boldsymbol{y}_{1:T}, \boldsymbol{u}_{1:T})\right]$$

Using the reparameterization trick, we achieve an empirical estimate of the gradient of ELBO using samples from an independent noise distribution.

$$\boldsymbol{x}_{1:T}^{(k)} = h_{\boldsymbol{\Phi}}(\boldsymbol{\eta}; \boldsymbol{y}_{1:T}^{(k)}, \boldsymbol{u}_{1:T}^{(k)}); \quad \boldsymbol{\eta}_{nt}^{(k)} \sim \mathcal{N}(0, 1)$$
$$\nabla_{\boldsymbol{\Phi}, \boldsymbol{\Theta}}\mathcal{L}(\boldsymbol{\Phi}, \boldsymbol{\Theta}) = \mathbb{E}_{\boldsymbol{\eta}}\left[\nabla_{\boldsymbol{\Phi}, \boldsymbol{\Theta}}\log p_{\boldsymbol{\Theta}}(h_{\boldsymbol{\Phi}}(\boldsymbol{\eta}; \boldsymbol{y}_{1:T}, \boldsymbol{u}_{1:T}), \boldsymbol{y}_{1:T}, \boldsymbol{u}_{1:T}) - \log q_{\boldsymbol{\Phi}}(h_{\boldsymbol{\Phi}}(\boldsymbol{\eta}; \boldsymbol{y}_{1:T}, \boldsymbol{u}_{1:T}))\right]$$

In practice, we use Monte Carlo estimates of the expectation above by sampling from an independent noise distribution and evaluating the gradient inside the expectation for those samples. The number of samples used for estimating the gradient introduces a trade-off between the accuracy of the estimation computation time. We empirically find that a single sample is often enough to achieve accurate estimates. We choose `Adam` optimizer to perform our stochastic gradient-based optimization.

Crucially, the function $h_{\boldsymbol{\Phi}}$ which takes in the data at each trial and combines it with independent noise to generate the latent trajectories is an LSTM. We choose LSTM to match the data domain (i.e. time series) and ensure that the variational distribution respects the time causality while being expressive enough to fit our datasets. Specifically, we have the following decomposition of the variational distribution.

$$q_{\boldsymbol{\Phi}}(\boldsymbol{x}_{1:T}) = \prod_{t=1}^{T} \mathcal{N}\left(\boldsymbol{\mu}_{\boldsymbol{\Phi}}(\boldsymbol{y}_{1:t}, \boldsymbol{u}_{1:t}), \sigma_{\boldsymbol{\Phi}}(\boldsymbol{y}_{1:t}, \boldsymbol{u}_{1:t})\right)$$
$$h_{\boldsymbol{\Phi}}(\boldsymbol{\eta}; \boldsymbol{y}_{1:T}^{(k)}, \boldsymbol{u}_{1:T}^{(k)}) = \boldsymbol{\mu}_{\boldsymbol{\Phi}}(\boldsymbol{y}_{1:t}^{(k)}, \boldsymbol{u}_{1:t}^{(k)}) + \boldsymbol{\eta} \times \sigma_{\boldsymbol{\Phi}}(\boldsymbol{y}_{1:t}^{(k)}, \boldsymbol{u}_{1:t}^{(k)})$$

where both $\boldsymbol{\mu}_{\boldsymbol{\Phi}}, \sigma_{\boldsymbol{\Phi}}$ are LSTM functions and the inference is done in an amortized fashion. An important distinction in our variational inference scheme is the application of interventional inputs directly in the variational mean $\boldsymbol{\mu}_{1:T} := \boldsymbol{\mu}_{\boldsymbol{\Phi}}(\boldsymbol{y}_{1:T}, \boldsymbol{u}_{1:T})$. This is achieved via replacing the variational mean at time point $t$ denoted by $\boldsymbol{\mu}_t$ with its interventional counterpart $1\{\boldsymbol{B}\boldsymbol{u}_t = 0\} \otimes \boldsymbol{\mu}_t + \boldsymbol{B}\boldsymbol{u}_t$.

## C.2. Computational Complexity

In order to examine the computational complexity of our variational framework, we first expand the gradient calculation.

$$
\begin{aligned}
\nabla_{\boldsymbol{\Phi}, \boldsymbol{\Theta}} \mathcal{L}(\boldsymbol{\Phi}, \boldsymbol{\Theta}) &= \mathbb{E}_{\boldsymbol{\eta}} \left[ \nabla_{\boldsymbol{\Phi}, \boldsymbol{\Theta}} \log p_{\boldsymbol{\Theta}}(h_{\boldsymbol{\Phi}}(\boldsymbol{\eta}; \boldsymbol{y}_{1:T}, \boldsymbol{u}_{1:T}), \boldsymbol{y}_{1:T}, \boldsymbol{u}_{1:T}) - \log q_{\boldsymbol{\Phi}}(h_{\boldsymbol{\Phi}}(\boldsymbol{\eta}; \boldsymbol{y}_{1:T}, \boldsymbol{u}_{1:T})) \right] \\
&\approx \sum_{k} \nabla_{\boldsymbol{\Phi}, \boldsymbol{\Theta}} \log p_{\boldsymbol{\Theta}}(h_{\boldsymbol{\Phi}}(\tilde{\boldsymbol{\eta}}; \boldsymbol{y}_{1:T}^{(k)}, \boldsymbol{u}_{1:T}^{(k)}), \boldsymbol{y}_{1:T}^{(k)}, \boldsymbol{u}_{1:T}^{(k)}) - \nabla_{\boldsymbol{\Phi}, \boldsymbol{\Theta}} \log q_{\boldsymbol{\Phi}}(h_{\boldsymbol{\Phi}}(\tilde{\boldsymbol{\eta}}; \boldsymbol{y}_{1:T}^{(k)}, \boldsymbol{u}_{1:T}^{(k)}))
\end{aligned}
$$

where $\tilde{\boldsymbol{\eta}}$ is a sample from i.i.d gaussian noise. The estimator above involves (1) generating $h_{\boldsymbol{\Phi}}$ by running two LSTM forward calls and running backprop to compute the gradients wrt $\boldsymbol{\Phi}$; (2) passing the generated latent trajectory to the generative model with linear dynamics and nonlinear emissions and computing gradients of the log probability w.r.t the parameters $\boldsymbol{\Theta}$. Both of these scale linearly with time $T$, and quadratically with the latent dimension $D$ and $N$. In addition, running backprop on the LSTM functions as well as the emission function depends on the size of the architecture used to realize those functions.

