# OpenReview forum: "Identifying Neural Dynamics Using Interventional State Space Models"
_ICML.cc/2025/Conference — ICML 2025 poster_

### Official Review · Reviewer_KJn6 · 2025-03-09

**Overall Recommendation:** 4

**Summary:**

The paper introduces interventional state space models (iSSM), a novel framework designed to predict neural responses to novel perturbations, addressing the limitations of traditional state space models (SSM) which capture statistical associations without causal interpretation. The authors establish the identifiability of iSSM. The model is validated through simulations of the motor cortex and applied to biological datasets, including calcium recordings from mouse ALM neurons and electrophysiological recordings from macaque dlPFC. The results demonstrate that iSSM outperforms traditional SSMs in terms of parameter identifiability and predictive power.

## update after rebuttal
Thanks for the detailed response and the additional sections on reproducibility, inference, and computational complexity. I also appreciate the short discussion on the optimal latent dimension. These changes address my feedback and strengthen the paper. My evaluation remains positive, and I’ll keep my original score of 4 (Accept) unchanged. I encourage the authors to further explore testing the optimal latent dimension in future work.

**Claims And Evidence:**

Yes

**Essential References Not Discussed:**

-

**Experimental Designs Or Analyses:**

Yes

**Methods And Evaluation Criteria:**

Yes

**Other Comments Or Suggestions:**

-

**Other Strengths And Weaknesses:**

## Strengths
- Theoretical Contribution: The paper provides a theoretical foundation for the identifiability of iSSM, which is crucial for causal inference in general especially when the goal is to design some sort of control strategy to steer the system towards a desired state.
- Empirical Validation: The model is tested on both synthetic and real-world datasets, showing generalization capabilities and parameter recovery.
- Practical Relevance: The application of iSSM to real biological datasets demonstrates its potential utility in neuroscience research, particularly in understanding neural dynamics under perturbations. Although the current state of technology may not allow a granular intervention on latent dimensions, the result of this work could be of higher practical use when such interventional capabilities become possible.

## Weaknesses
- Nonlinearity Limitation: The current model focuses on linear dynamics, and while the inference model can capture nonlinearities, explicitly modeling nonlinear dynamics in the latent space remains a limitation.
- Complexity of Implementation: The implementation details and computational complexity of iSSM are not thoroughly discussed, which could be a barrier for adoption by other researchers.
- Limited Scope of Interventions: The paper primarily assumes that interventions force the neural state out of its low-dimensional attractor manifold. I think this assumption and its consequences need to be deeper understood as they seem to be critical for the results of this work.

**Questions For Authors:**

- Although the emission model introduces nonlinearity to the model, the dynamics in the latent space is linear and cannot express qualitative behaviours that only nonlinear dynamical systems may have, e.g. limit cycles. This might be a guide for where not to use this model. Can you provide examples of neural circuits for which the linear dynamics is insufficient to exhibit the underlying dynamics regardless of how expressive the emission model is?
- It seems the latent dimension is crucial for this model as those dimensions are supposed to correspond to biologically meaningful behaviour and to be physically intervenable. The stated observation in the paper “increasing the number of latents always provides better test accuracy” seems counterintuitive as there should be a number of dimensions that corresponds to the physical properties of the system and beyond that there is a risk of overfitting. Could you elaborate on this observation and any reasoning behind it?
- I recommend including a pseudocode in the appendix and provide a more detailed discussion of the implementation. An ablation study would also be useful to better understand the author’s design choice, e.g. why choosing LSTM as the recognition network over other options. The authors have mentioned they have done cross-validation to find the optimal set of hyper-parameters, but it’s unclear which hyperparameters were involved in that search and with which range. Also how the range of search for hyper-parameters changes with the problem at hand.

**Relation To Broader Scientific Literature:**

The paper could be related to causal discovery and control of dynamical systems.

**Theoretical Claims:**

I checked the proof of Theorem 3.4. It seems correct.

---

> ### Author Rebuttal · Authors · 2025-04-01
>
> We sincerely appreciate you and the other reviewers for your time and thoughtful evaluation of our work. We found the feedback to be highly constructive, as well as both fair and encouraging. Below we address your specific questions:
>
> **Nonlinearity limitation:** We completely agree with the reviewer that linear dynamics introduce a limitation into the model. However, developing theoretical results for more complex models is quite challenging. To our knowledge this is the first paper that develops such results for a non-LDS noisy SSM. We hope this brings the causal inference and neuroscience communities closer together and inspires the development of more theoretical results for more complex models.
>
> **Complexity of implementation:** Thanks for pointing this out, we have included a new section in the supplementary material detailing the inference and its computational complexity. To summarize the section we find that our algorithm scales as $\mathcal{O}(TN^2 + TD^2)$ where $T$ is the length of the trajectories, $D$ is the latent dimension, and $N$ is the number of neurons. Therefore our inference is linear in time and quadratic in the number of neurons.
>
> **Limited score of interventions:** The existing evidence for interventions forcing the neural state outside of the observational trajectories is discussed in both prior theoretical and experimental papers [1, 2, 3]. That said, we believe that collectively as a field our understanding of how interventions affect the neural states is quite limited. The model we present here is among the first models that incorporate interventions into the modeling framework and therefore we believe it’s critical to develop interventional models to better understand how interventions influence neural dynamics.
>
> [1] Jazayeri, M. and Afraz, A. Navigating the neural space in search of the neural code. Neuron, 93(5):1003–1014, 2017.
>
> [2] O’Shea, D. J., Duncker, L., Goo, W., Sun, X., Vyas, S., Trautmann, E. M., Diester, I., Ramakrishnan, C., Deisseroth, K., Sahani, M., et al. Direct neural perturbations reveal a dynamical mechanism for robust computation. bioRxiv, pp. 2022–12, 2022.
>
> [3] Daie, K., Svoboda, K., and Druckmann, S. Targeted photostimulation uncovers circuit motifs supporting short-term memory. Nature neuroscience, 24(2):259–265, 2021.
>
> **Expressivity of the emissions:** While we agree that linear dynamics is a limiting factor of the model, we would like to point out that adding nonlinear emissions to the linear dynamics changes its expressivity quite drastically. We reference the Koopman theory in the paper, which posits that a (possibly infinite dimensional) linear dynamical system followed by a nonlinear emission is sufficient to express "any" autonomous dynamical system. The catch here is of course "infinite dimensional" but intuitively as the latent dimension grows larger the expressivity of the model also increases.
>
> **Increasing latent dimension:** You are absolutely correct that if we assume low-dimensionality of the data then increasing the latent dimension is counter-intuitive. We would like to clarify that while we think the optimal latent dimension crucially depends on the task and dataset, we think of interventions as kicking the state of the system outside of its low-dimensional manifold increasing the dimensions spanned by the trajectories. Therefore the conjecture we've included in the discussion section is only valid for large datasets where diverse and numerous interventions are performed. But we have only presented this argument as a conjecture and we agree that experimental validation is needed for confirming it.
>
> **Discussion of the implementation:** Thank you for the suggestion, we added a new section in the supplementary material on the details of our variational inference algorithm. We have also included our code package in the submission which follows standard code development practices.
>
> **LSTMs, hyperparameters:**
> Using LSTMs in the variational posterior has become a standard practice in the field (e.g. [1]). We did experiment with vanilla RNNs but they drastically underperformed LSTMs. We will include these results if the paper gets accepted. Regarding the hyperparameter search, our two hyperparameters are the sparsity of $\boldsymbol{B}$ denoted by $s$ and the latent dimension $D$. The cross-validation and the range of parameters are included in Fig. 3C, 4D. We used a logarithmic axis for s to cover a wide range of possibilities and for $D$ we increased it up to a point where we see a drop of validation performance. Regarding how the hyperparameters change w.r.t. the task at hand, this question is highly problem specific. We think these value highly affected by the details of the experimental setup such as the task, type of recording, animal model, etc.
>
> [1] Krishnan, Rahul G., Uri Shalit, and David Sontag. "Deep Kalman Filters." arXiv preprint arXiv:1511.05121 (2015).
>
> Thanks again for your time; we look forward to your final assessment.

---

> > ### Comment · Reviewer_KJn6 · 2025-04-07
> >
> > Thank you for the detailed response. I appreciate the authors’ efforts in addressing my questions and incorporating clarifications and additional material into the revision. I find this work to be a valuable contribution to the community, and my overall assessment remains positive. As my initial score already reflects this opinion, I will keep it unchanged. I encourage the authors to expand on the reasoning behind their conjecture regarding the optimal latent dimension and suggest concrete ways it could be tested in future work.

---

> > > ### Author Response · Authors · 2025-04-08
> > >
> > > Thank you for the encouraging words. We have included new sections in our appendix on (1) reproducibility details (2) inference details (3) computational complexity. In addition, we will include a short discussion on the optimal dimension. We appreciate your time and feedback.

---

### Official Review · Reviewer_kBw8 · 2025-03-12

**Overall Recommendation:** 3

**Summary:**

The paper introduces Interventional State Space Models (iSSM), a class of causal state-space models designed to identify neural circuit dynamics and predict responses to causal manipulations, addressing the limitations of traditional state-space models (SSMs) which lack causal interpretability. By explicitly modeling perturbations as interventions on latent neural dynamics, the authors demonstrate theoretically that iSSMs achieve identifiability of latent dynamics under suitable conditions. Empirically, they validate their model using simulated motor cortex dynamics and biological datasets involving calcium imaging from mouse ALM neurons during targeted photostimulation, and electrophysiological recordings from macaque dlPFC during micro-stimulation. Across these examples, iSSM consistently outperforms conventional observational SSMs in reconstructing neural activity, demonstrates robustness to initialization, and effectively generalizes to predict neural responses under novel interventions.

**Claims And Evidence:**

The main claims are well-supported by both theoretical arguments and empirical results.

**Essential References Not Discussed:**

Yes.

**Experimental Designs Or Analyses:**

Yes. I have the following questions:
- In Figure 3.B, what's the difference of latent activities between correct and incorrect trials?
- The goal of Section 4.3 is to evaluate iSSM's ability to generalizetounseeninterventions. Could you explain more about how this is achieved via Figure 4?

**Methods And Evaluation Criteria:**

Yes, the paper clearly demonstrates alignment between the proposed methods, evaluation approaches, and the key problems it addresses. The authors provide theoretical guarantees showing that iSSMs achieve identifiability under suitable assumptions (e.g., bounded completeness, piecewise linearity, sufficient interventions), thus enabling accurate predictions under novel perturbations. They empirically validate these claims using synthetic motor cortex simulations and biological datasets (calcium imaging in mice and electrophysiology in macaques), showing that iSSM consistently outperforms standard observational models (SSM) in reconstructing neural activity, recovering latent dynamics, and ensuring robustness of learned parameters across initializations.

**Other Comments Or Suggestions:**

N/A

**Other Strengths And Weaknesses:**

Strengths:
- Combining causal inference concepts with traditional state-space modeling techniques, bridging a critical gap between purely statistical and causal modeling approaches.
- Advances theoretical understanding by providing formal identifiability conditions, clearly extending prior theoretical work to the specific and challenging domain of neural dynamics.

Weakness:
- Only a standard SSM is used as a baseline. The authors should consider including additional baseline models specifically designed for causal dynamical systems to better demonstrate the superiority of iSSM.

**Questions For Authors:**

N/A

**Relation To Broader Scientific Literature:**

The paper’s key contributions build upon and directly address limitations highlighted in prior literature: traditional SSMs, effectively capture statistical correlations but fail to offer causal interpretability.  This work specifically incorporates theoretical insights on the identifiability of nonlinear causal dynamical systems and extends them by allowing observational noise, a crucial step for realistic biological modeling

**Theoretical Claims:**

Yes.

---

> ### Author Rebuttal · Authors · 2025-04-01
>
> We sincerely appreciate you and the other reviewers for your time and thoughtful evaluation of our work. We found the feedback to be highly constructive, as well as both fair and encouraging. Below we address your specific questions:
>
> **Correct vs. incorrect trials:** Please notice the latents closer to the end of the trial where the correct and incorrect latents start to diverge. We noticed that that’s the time when the animal is performing its action selection. This is relevant for us because the area where the recordings are performed ALM is thought to involve action selection. Therefore our results suggest that the causal latents discovered by the model distinguish between the action selection mechanism for the correct vs. incorrect trials.
>
> **Generalization to unseen interventions:** As described in the last paragraph of section 4.3, we divide the data into train and test trials with different interventions performed in each set. Our results show that (Fig. 4 C,D,E) that the test reconstruction accuracy is on par with training reconstruction accuracy, supporting the argument that the model is able to generalize to test interventions.
>
>
> **Compare against other baselines:** Thank you for pointing at the weakness of the paper. We want to clarify that our contribution is not developing a new latent variable model, rather it’s adding the interventional components to the existing SSMs. To our best knowledge, this is the first causal dynamical systems model that incorporates latents and observational noise. Therefore we are not aware of other alternatives to use for our comparisons, we are happy to include new comparisons if the reviewer has specific methods in mind.
>
> Thanks again for your time; we look forward to your final assessment.
>
> &nbsp;
>
> ### **Further responses to reviewer xGnX**
> **Explanation of dynamic attractor:**  We borrowed the term "dynamic attractor" from the previous literature on this toy example [1]. In our understanding, a ring attractor in the neuroscience literature implies a specific latent mechanism where the spatial information is encoded in the connectivity pattern of the network. Here the dynamic attractor has non-rotational latents that are nonlinearly transformed to generate rotational dynamics in the observational space. Indeed the dynamical structure generated in the observational space is a ring attractor but we remained consistent with the terms in the literature.
>
> Indeed you are correct that in the non-noisy setting $\boldsymbol{x}=(0,1)$ or equivalently $\boldsymbol{y}=(0,0)$ is an unstable fixed point of the model. So the model will not move away from it. However, except for that pathological case, all other settings of $\boldsymbol{x}$; $\frac{d\boldsymbol{x}}{dt}$ will be nonzero and the model does not require perturbation to move along the circle.
>
> [1] Galgali, A. R., Sahani, M., and Mante, V. Residual dynamics resolves recurrent contributions to neural computation. Nature Neuroscience, 26(2):326–338, 2023.
>
> **Positive observations in Fig. 2C**: Thank you for your observation. This is a plotting issue, and the y-axis in the plots is not meaningful. We will fix that in the final revision (if accepted). We min-subtract the data in each dimension to stack different axes of the latents on top of each other. What’s shown in the state plots (Fig. 2 A, B) shows the exact numbers corresponding to the trajectories.
>
> **Sparsity penalty on $\boldsymbol{B}$**: We mention in line 132-134 (right column) that the sparsity penalty is implemented by placing a Laplace prior on the elements of $\boldsymbol{B}$ with a scale parameter $s$. In our real data experiments we do cross-validation over $s$ (the columns of Fig. 3C and Fig. 4D). For synthetic data we don’t use a sparsity prior.
>
> **Injective $f$**: While we don't explicitly enforce that the function f is an injective function, this function maps from a low-dimensional latent space for x to a high-dimensional observational space. So if this function fits the data well, then this function is often injective without explicitly enforcing it, as long as none of the latent dimensions collapses and are not used --- namely many different observations are mapped to the same latent variable value. We can often empirically check that this issue does not occur.
>
> **Reconstruction accuracy:** The reconstruction accuracy is defined as the correlation coefficient between true and reconstructed signals (latents or observations). Notice that except for the case of stimulation count is 0, the observational reconstruction accuracy is always close to 1 for both models. However, the latent reconstruction accuracy which directly measures the identifiability of the latents is only close to one for iSSM as the stimulation count increases. This is the main signature that the added interventional component to the model indeed leads to identifiability. The latent dimension for all the results in this figure is set to the ground truth 2.

---

> > ### Comment · Reviewer_kBw8 · 2025-04-06
> >
> > Thanks for the clarification. I will keep my score.

---

> > > ### Author Response · Authors · 2025-04-08
> > >
> > > Thank you for your time, please feel free to check our new results included in response to reviewer **xGnX**. We hope our new results improve the quality of the paper.

---

### Official Review · Reviewer_xGnX · 2025-03-12

**Overall Recommendation:** 1

**Summary:**

The authors propose iSSM, which is a linear dynamical systems model that accounts for causal perturbations to the neural population activity. The authors apply this model to two synthetic datasets inspired by literature on motor cortex dynamics, and apply their model to a variety of real neural population datasets that include perturbations.

**Claims And Evidence:**

The authors present theoretical results for the identifiability of iSSM. The authors claim that the “framework is general and can be applied to different types of SSMs, in this paper we focus on adding interventional components to the SSMs with linear dynamics and nonlinear observations.” However, I don’t think there was enough information in the paper to conclude to what extent the authors’ results apply. For example, can we replace the dynamics matrix A with a neural network and would the results still hold? How about if both matrix A and B are replaced with a neural network such that we have a completely nonlinear dynamics x_{t+1} = g(x_t, u_t), where g is an MLP?

I think the authors shouldn’t say persistent activity is a hallmark of short-term memory (Line 325), especially when they reference Goldman, 2009. The idea in Goldman, 2009 is precisely that we can memorize without persistent neural activity.

**Essential References Not Discussed:**

I can’t think of essential references not cited.

**Experimental Designs Or Analyses:**

It was not clear how reconstruction accuracy was calculated in Figure 2. I am also surprised that using this metric, reconstruction accuracy is close to 0 for both iSSM and SSM in Figure 2D. It was not clear how the error bar is computed. For this analysis, are the authors assuming that the latent dimensionality is 2?

I think it is concerning that there is not enough details in the experiments to reproduce the results. For example, for the real datasets, how was the latent dimensionality chosen? What was the hyperparameter space? How was the hyperparameter optimized? How was the training and test split made? When the authors mention they do cross-validation, how many folds? (e.g., “To find the optimal hyperparameters, we performed cross-validation.” (Line 379) How many folds were used? What fold was used for showing the result in Fig 4D?)

Consistency score is defined only with respect to B. Have the authors evaluated consistency for A?

In Figure 3, the mean latents for correct and incorrect trials seem very similar. Given that the reconstruction accuracies for both SSM and iSSM are less than ~0.5, I’m not sure if the latent trajectories here should be trusted. To have some confidence, I think it would make sense to fit this dataset to an established model such as LDS with linear observations to see how SSM and iSSM perform with respect to this LDS baseline.

Is data relevant to Figure 4 spike trains? It is odd that the plots in Figure 4A are in terms of firing rates, and not binned spike counts. Do the authors assume Poisson observations or Gaussian?

**Methods And Evaluation Criteria:**

I couldn’t find sufficient details of how inference was done in the model, either in the main text or the Appendix. For example, having in the Appendix the details of the LSTM architecture, how the number of units in the LSTM was chosen, whether it is bidirectional or unidirectional, and whether it is an LSTM that runs forward in time or backward in time would be helpful. I also couldn’t find sufficient details on how the synthetic datasets were generated. For example, it was not clear what a_1 and a_2 were set to be. How many trajectories were generated for this synthetic dataset? What was the initial condition of the dynamics? How was the dynamical system perturbed in this simulation? Without these details (sufficient enough for the readers to reproduce the authors’ results), it is difficult to evaluate the authors’ claim.

The authors claim that in the synthetic datasets, iSSM can recover the true latents and the underlying dynamics. However, the analyses didn’t have important details. For example, the authors show gray arrows (flow fields) in Figure 2A and B. Are the arrows both from the ground truth data or are they model-inferred? The flow fields for true and inferred seem identical. A clarification on this would be helpful. I would like to see how well iSSM recovers the flow fields, not just the trajectories.

It was also not mentioned in the paper (both main text and Appendix) how SSM is defined. I think the definition of what SSM is should be in the main text. Is it a linear dynamical system (LDS) model with nonlinear observations similar to Gao et al., 2016? What inference method was used for SSM? Is it similar to the inference procedure for iSSM?

Furthermore, I am uncertain whether this is the right type of synthetic datasets because it seems that the number of observations here is only two, whereas in the real datasets that the authors look at, the number of neurons tends to be higher.

Is “Dynamic Attractor” a ring attractor? The flow field shown in Figure 2A1 and 2B1 seem to suggest this. However, if this is a ring attractor, I am confused because in Equation (6), dx/dt = 0 if and only if x_1 = 0 and x_2 = 1. This means that y_t = [cos(1); 0] for all t. However, the trajectory y_t is not a circle, contrary to what is shown in Figure 1A1 and 2B1. I think it is not clear to me how the flow field plots are generated. For “Dynamic Attractor”, is there a way for the neural state to move without perturbations? Rotational dynamics are seen in neural data even when there are no optogenetic or electric stimulations.

In Figure 2B, the observation states are sometimes negative, but in Figure 2C the observations and latents are always positive. Is the trajectory in Figure 2A-B not the same as C? A clarification on how many trajectories are generated in total in this dataset would be helpful here as well.

It was not mentioned in the paper how B with sparsity penalty was implemented. How was the scale parameter s chosen? What was s in the experiments?

**Other Comments Or Suggestions:**

I have no other comments or suggestions. The paper was written clearly.

## update after rebuttal

I sincerely thank the authors for the detailed response. I could access the link and look at the details on estimated flow fields, 6d synthetic experiments, and details on initialisation, hyperparameters and inference (though I believe full details on the inference should be available in the Appendix for this to be published).

**Dynamic Attractor:** if the equation should be $\frac{dx}{dt} = [a_1x_1, a_2(1-x_1)]^\top$ instead of $\frac{dx}{dt} = [a_1x_1, a_2(1-x_2)]^\top$, then I don't think there is a fixed point in this system if $a_1 = -20$ and $a_2 = 1.2$ as stated in the shared link. This is inconsistent with the authors' claim that the system is a dynamic attractor.

**Sparsity on B:** I re-read the Appendix of the original submission and could not find information about the loss function. The loss shows up in 3.2, but it doesn't have any details on regularization. The shared link says the regularization coefficient was found using cross-validation, but no more details could be found.

**Rotational Dynamics:** Should we be worried that the red flow field looks more like a line attractor, whereas for the black arrows, there doesn't seem to be a line of fixed points? Have the authors compared the eigenvalues?

**Reconstruction accuracy:** It makes sense that if the latents are not identifiable, the correlation could be close to 0 for the latents (D1), but it is still puzzling to me why the neural reconstruction (D2) is also close to 0. This is also partly why I think having benchmarks to models like LDS (with linear emission) could be helpful. Does the LDS show that the neural reconstruction accuracy is close to 0?

These remaining unresolved questions raise concerns about the validity of the analysis, and I will keep my score as is.

**Other Strengths And Weaknesses:**

Please see above.

**Questions For Authors:**

1. The authors mention that “models that are built upon observational data are not able to capture neural dynamics outside of the low-dimensional space.” However, it is not clear to me whether this method can model dynamics that are outside the latent x. How do the authors model dynamics that are outside the intrinsic manifold?

2. Can iSSM model inputs that are not interventional as well? If so, how does that affect its performance in comparison to SSM?

3. Could we have used something other than LSTM for the posterior? I wonder if there are any alternative ways of doing inference.

**Relation To Broader Scientific Literature:**

A major strength of the paper is that the idea of incorporating interventional data into SSMs is interesting, and, if it works, would be an important contribution to the field. As the authors have pointed out, inferring latent dynamics from neural population activity is difficult, and if we have interventional data that can further constrain the model to learn the correct dynamics, this would help us understand neural computation. However, the experimental evidence supporting this idea is weak.

**Theoretical Claims:**

I did not check the proofs of the theoretical claims. However, I would like to note that one of the assumptions is that the function f is an injective function. It was not clear how this was enforced during training of the authors’ model.

---

> ### Author Rebuttal · Authors · 2025-04-01
>
> We sincerely appreciate you and the other reviewers for your time and thoughtful evaluation of our work. We found the feedback to be highly constructive, as well as both fair and encouraging. Below we address your specific questions:
>
> **Generalizing to other SSMs:** Thank you for bringing this up. Indeed we present our theoretical results for linear dynamics and nonlinear observations. The algorithmic part of the paper is readily applicable to more complex generative models. Since we’re using variational inference which is agnostic to the underlying model, changing the components of the generative model–i.e. dynamics, emissions, and noise model–will not impact the inference. For identifiability results, the key idea of the proof lies in independence testing: The intervened latents at time t+1 are independent of the others at previous time step t, and this independence leads to the identifiability of the latents. The linearity assumption in the latent space does make this argument easy to make, since we only need to consider the covariance. That said, this argument is readily generalizable to other settings where $x_{t+1}$ and $x_{t}$ are nonlinearly related (e.g. polynomial function or ReLU function up to sparsity constraints), as long as the statistical independence of the independent latents is still sufficient to identify the latents. That said, our goal for this paper is laying the foundation of applying interventional models to neural datasets and we leave these developments for future work.
>
> **Hallmark of short-term memory:** We thank the reviewer for noticing this. While we agree that multiple mechanisms are proposed for working memory, here we’re using "persistent activity" loosely without any implications of the underlying mechanism. Notice that even in Goldman 2009 the proposed mechanism still explains the observed persistent activity. We will clarify in the revised manuscript that the observed persistent activity should not be confused with the underlying mechanism generating it.
>
> **Reproducibility:** Thank you for pointing this out. While we have not detailed the specifics of our inference scheme, we’re using standard variational inference tools. That said, we added a new section to the supplementary about the details of inference and computational complexity for completeness. Additionally, we realize that we’ve omitted some important reproducibility details, therefore we added a new section to the supplementary on this. Table [1](https://f.uguu.se/qDwlATSB.png), [2](https://d.uguu.se/jCNcSZzj.png) summarize these results.
>
> **Recovering true latents:** Thank you for this great observation. The arrows presented are all from the ground truth model, we have not shown the inferred dynamics by the model. We will include this in the revised manuscript. That said, notice that it’s not quite straightforward to visualize the inferred dynamics. The dynamics inferred by the model are encoded in the variational LSTM and the fitted emission. There are two ways for visualizing the inferred dynamics. (1) Using the inferred A: This is not ideal because of the variational gap. (2) Using the emission & variational posterior. This approach requires us to generate a grid in the observation space. However, due to the use of LSTMs the variational posterior is context dependent. We’re open to the reviewer’s suggestions for visualizing the dynamics, but so far we haven’t found an appropriate visualization.
>
>
> **Definition of iSSM:** The SSM and iSSM in all experiments share the exact same characteristics–i.e. same latent dimension, emission & inference architecture, step size, etc.–except for the interventional components where SSM models interventions as an additive term. We did this to focus on the utility of interventional models while accounting for all other confounding factors and bringing the two models to the same footing. In the codes, there is an `interventional: bool` flag which if set to `False` will model the interventions in an additive way.  For the Poisson experiment (Fig. 4) the model is equivalent to Gao et. al. 2016, except that they did not include any inputs to the model. For the Normal experiment (Fig. 2, 3) the model departs from Gao et. al. 2016 in terms of the observational noise model.
>
> **Low-d synthetic experiments:** The synthetic experiment in Fig. 2 is mainly presented to provide intuition. It’s an example where latents are clearly defined from a behavioral standpoint and the nonlinear emission is required to produce rotational trajectories. We think it’s an intuitive example for the computational neuroscience audience because it’s a toy version of two prevalent hypotheses about rotational dynamics in the motor cortex. The synthetic example in Supp. Fig. 7 uses higher dimensional latents.
>
> We used the remaining space in the responses to reviewer **3PQB** and **kBw8** to address your remaining concerns. Thanks again for your time; we look forward to your final assessment.

---

> > ### Comment · Reviewer_xGnX · 2025-04-05
> >
> > I thank the authors for the detailed response.
> >
> > **Generalizing to other SSMs:** Thank you for clarifying that the theoretical results only hold for linear dynamics.
> >
> > **Hallmark of short-term memory:** Thank you for the clarification. This makes sense.
> >
> > **Reproducibility:** I was not able to access Table 1, 2 on my end. Could you please share different links?
> >
> > **Recovering true latents:** Could the authors elaborate on what they mean by the variational gap and how the variational posterior is context-dependent? Without full details on the inference method and precise definitions of the terms, it is hard to evaluate correctness, but based on the info that I currently have, I think visualizing inferred dynamics using (1) is a valid way and more standard than (2). Some quantitative metric showing that the true and inferred flow fields match would be necessary for publication.
> >
> > **Definition of SSM:** The definition of SSM makes sense in that they are equivalent to iSSM, except for modeling interventions as additive. However, I’m puzzled by the authors’ statement that it is equivalent to Gao et al. for Fig. 4. If this was so, the authors should state this in the paper. Gao et al. doesn’t use LSTMs for the posterior whereas the authors’ model does, so they are not exactly equivalent.
> >
> > **Low-d synthetic experiments:** What about low-dimensional latents with large number of neurons? This is the kind of synthetic datasets used for validating e.g., LDS, rSLDS and LFADS in literature, so having them would be great.
> >
> > **Dynamic Attractor:** The Dynamic Attractor in Equation (S8) of [1] can generate a ring attractor because (S8) is in polar coordinates. For Equation (6), I don’t think the transformation gives a ring attractor, but a point attractor, as pointed out in my previous review. I think my questions above regarding what a_1 and a_2  were missed by the authors. I’m puzzled by how it is possible to generate flow fields like in Figure 2, which is a dynamic attractor like (S8) of [1] with the authors’ Equation (6). If it is indeed the case that dx/dt is non-zero at points other than the origin and the model doesn’t require perturbations to move along the circle, then I think this is inconsistent with the flow field that the authors plot.
> >
> > **Positive observations in Fig. 2C:** If what the authors say is true, is it possible to learn the correct flow field with this low number of trajectories?
> >
> > **Sparsity on B:** Thank you for the clarification. Does this mean there is an l1 regularization term in the loss function? Having more details in the supplement would be great.
> >
> > **Injective f:** Yes, I agree with the authors that often f can be injective without enforcing it to be injective. The authors mention they can check this doesn’t occur, but have the authors verified that the for models shown in the results, the learned f is injective?
> >
> > **Reconstruction accuracy:** Is it the correlation coefficient and not the coefficient of determination?
> >
> > **Question 3:** The authors did not address my question of whether the LSTM is running forward/backward, what the size of the LSTM is, etc. Could the authors provide references to back up the claim that “Using LSTMs as the variational family has become standard in the literature of SSMs”?
> >
> > The authors’ response did not answer why for stimulus count of 0, the reconstruction accuracy is so low for both SSM and iSSM.
> >
> > Other questions that I had in my original review below were not sufficiently addressed by the authors:
> >
> > >I think it is concerning that there is not enough details in the experiments to reproduce the results. For example, for the real datasets, how was the latent dimensionality chosen? What was the hyperparameter space? How was the hyperparameter optimized? How was the training and test split made? When the authors mention they do cross-validation, how many folds? (e.g., “To find the optimal hyperparameters, we performed cross-validation.” (Line 379) How many folds were used? What fold was used for showing the result in Fig 4D?)
> >
> > >Consistency score is defined only with respect to B. Have the authors evaluated consistency for A?
> >
> > >In Figure 3, the mean latents for correct and incorrect trials seem very similar. Given that the reconstruction accuracies for both SSM and iSSM are less than ~0.5, I’m not sure if the latent trajectories here should be trusted. To have some confidence, I think it would make sense to fit this dataset to an established model such as LDS with linear observations to see how SSM and iSSM perform with respect to this LDS baseline.
> >
> > >It is odd that the plots in Figure 4A are in terms of firing rates, and not binned spike counts... Could the authors clarify why this is in firing rates?
> >
> > The authors’ response raises further concerns about the validity of the analyses and rigour, and I believe the paper as is does not meet the standards for publication. I adjusted my score to reflect this.

---

> > > ### Author Response · Authors · 2025-04-08
> > >
> > > Thank you for engaging in the conversation, below we include the answers to your additional questions.
> > >
> > > **Table 1, 2**: We apologize for this, the links were generated by a free website and were expired short after. Here's the new [link](https://anonymous.4open.science/r/issm-figs-A800/repro.pdf) to both tables.
> > >
> > > **Recovering true latents**: We included the flow fields in the latent space [here](https://anonymous.4open.science/r/issm-figs-A800/flow-fields-x.png). While the fields aren't perfectly recovered but the general direction of the arrows are consistent.
> > >
> > > **Definition of SSM**: The original question was misunderstood as asking whether the generative models of iSSM and PfLDS are equivalent under certain configurations. When iSSM uses Poisson observations and excludes inputs, its generative model matches that of Gao et al's PfLDS. However, their inference methods and emission architectures differ. In the experiments, all factors were controlled using our implemented codebase, isolating the comparison to interventional versus additive inputs.
> > >
> > > **Low-d synthetic experiments**: We have included a new [figure](https://anonymous.4open.science/r/issm-figs-A800/motor-6d.png) in the rebuttal which projects the latent dynamics in the synthetic motor experiment into higher dimensions (6 dimensions). We will include more elaborate high-dimensional results in the appendix if we get accepted.
> > >
> > > **Dynamic Attractor** : We identified a small typo in the equation for Dynamic Attractor. Indeed the equation should be fixed to $\frac{dx}{dt} = [a_1 x_1, a_2 (1-x_1)]^{\top}$. All our results and flow fields are based on the correct equation, in fact the flow fields are programmatically generated by taking a grid and transforming according to the dynamics.
> > >
> > > **Sparsity on B**: Indeed an independent Laplace prior on the elements of $B$ translates into $l_1$ regularization on the matrix elements when computing log likelihoods. The information about the loss function is already included in the supplementary, we're happy to include a summary of the inference if the reviewer requests.
> > >
> > > **Injective f**: The injectivity of $f$ requires different $x_t$'s are mapped to different $f(x_t)$'s. We check it by checking the gradient of the learned $f(.)$ is nonzero almost surely. Moreover, the identification of the latents in synthetic experiments suggests that the map is indeed injective.
> > >
> > > **Reconstruction accuracy**: This is correct, we have included references in response to reviewer **3PQB** on the usage of this in the prior literature, please check our response above.
> > >
> > > **Question 3**: The LSTM always runs forward. The size of the LSTM is included in the new reproducibility tables linked above.
> > >
> > > **Stimulus count of 0**: We had to skip some questions due to character limits. When stim count is nonzero, the observational reconstruction accuracy is always close to 1 for both models. However, the latent reconstruction accuracy which directly measures the identifiability of the latents is only close to one for iSSM as the stim count increases. This is the main signature that the added interventional component to the model indeed leads to identifiability. When stim count is zero, the model does not have access to interventional data, therefore is not able to identify latents.
> > >
> > > **Reproducibility information**: We have included the reproducibility information in the tables, please let us know if you have further concerns.
> > >
> > > **Consistency score for A**: We only computed the consistency score to assess the robustness of the latent to perturbation cite relationships. Since the perturbation cite IDs are fixed we only need to account for the permutations of the latents by aligning the columns of $B$. However, to compute a consistency score for $A$ we need to permute the full matrix to account for permutation invariances, in the limited rebuttal time we were not able to perform this experiment, but if we get accepted we will include this in the final revision.
> > >
> > > **Latents for correct and incorrect trials**: Please notice the latents start to diverge close to the end of trial which hints at the type of information being encoded in the ALM region. The timing where the latents diverge corresponds to action selection which is thought to involve ALM. Therefore we believe the divergence of the signals is neuroscientifically meaningful and interesting. Regarding comparison with LDS, please notice that the model we’re fitting generalizes LDS. Therefore if LDS is a better fit to the dataset the inference should learn a simple linear emission.
> > >
> > > **Figure 4A firing rates**: We apologize for mislabeling the figure, the data shown in Fig. 4A is indeed spike counts and not firing rates.
> > >
> > > We hope our new results clarify our contributions and address your concerns. We believe our theoretical and algorithmic contributions are relevant to both neuroscience and causal dynamical systems communities and therefore we would appreciate your reconsideration of our score.

---

### Official Review · Reviewer_3PQB · 2025-03-14

**Overall Recommendation:** 3

**Summary:**

This paper provides an extension of the state space model (SSM) to an interventional SSM (iSSM). iSSM is able to causally identify and infer external inputs as an intervention to the neural dynamics, while also inferring the latent and reconstruct the observation accurately. Methods, assumptions, and derivations are clearly presented. These good properties are supported by one synthetic and three real-world experiments.

**Claims And Evidence:**

Yes.

**Essential References Not Discussed:**

/

**Experimental Designs Or Analyses:**

Yes.

**Methods And Evaluation Criteria:**

Yes.

**Other Comments Or Suggestions:**

/

**Other Strengths And Weaknesses:**

* The presentation is great.
* The experiment is comprehensive and scientifically interesting.

**Questions For Authors:**

* What is the definition of reconstruction **accuracy**. People usually use test likelihood, rather than accuracy, to evaluate observation data, at least in a lot of machine learning applications and especially the neuroscience field.
* Is the reconstruction accuracy based on the optimal latent $\boldsymbol x_t$ inferred by the variational distribution?
* Have the authors think more about the choice of variational distribution and its validity? LSTM is only unidirectional information flow, which might not be powerful enough to serve as the variational distribution. See, [this](https://proceedings.neurips.cc/paper/2007/hash/2bcab9d935d219641434683dd9d18a03-Abstract.html) and [this](https://openreview.net/pdf?id=2FKzbEE24s).
* Since the SSM model used in this paper is essentially a nonlinear LDS, there are a lot of LDS as mentioned at the beginning of this paper, such as SLDS. Is it possible to compare iSSM with SLDS? Comparing with the simple LDS seems a bit weak.

**Relation To Broader Scientific Literature:**

Related to a series of SSMs, which is a class of commonly used neural data and neural dynamics analysis tools in computational neuroscience.

**Theoretical Claims:**

Have checked in detail before Theorem 3.4.

---

> ### Author Rebuttal · Authors · 2025-04-01
>
> We sincerely appreciate you and the other reviewers for your time and thoughtful evaluation of our work. We found the feedback to be highly constructive, as well as both fair and encouraging. Below we address your specific questions:
>
> **Definition of reconstruction accuracy:** We noticed that the definition of the reconstruction accuracy is missing in the paper. The reconstruction accuracy is defined as the correlation coefficient between the true and reconstructed latents, which can only be computed for synthetic experiments with ground truth latents. It’s a common metric used in the causal representation learning literature for assessing the identifiability of the latents [1,2,3].
>
> [1] Khemakhem, I., Kingma, D., Monti, R., & Hyvarinen, A. (2020, June). Variational autoencoders and nonlinear ica: A unifying framework. In International conference on artificial intelligence and statistics (pp. 2207-2217). PMLR.
>
> [2] Khemakhem, I., Monti, R., Kingma, D., & Hyvarinen, A. (2020). Ice-beem: Identifiable conditional energy-based deep models based on nonlinear ica. Advances in Neural Information Processing Systems, 33, 12768-12778.
>
> [3] Song, X., Li, Z., Chen, G., Zheng, Y., Fan, Y., Dong, X., & Zhang, K. (2024). Causal temporal representation learning with nonstationary sparse transition. Advances in Neural Information Processing Systems, 37, 77098-77131.
>
>
> **Other variational distributions:** This is a great suggestion, we thank the reviewer for pointing this out. LSTMs have been used in previous SSM literature [1]. We tried vanilla RNNs and LSTMs and LSTMs were significantly better than RNNs. The performance enabled by LSTMs is already sufficient for identifiability experiments we show in the paper. Unfortunately due to the limited time we will not be able to provide new results using new variational families but if we benefit from new variational families we will incorporate the new results in the final manuscript upon acceptance.
>
> [1] Krishnan, Rahul G., Uri Shalit, and David Sontag. "Deep Kalman Filters." arXiv preprint arXiv:1511.05121 (2015).
>
> **Compare against other SSMs:** In all experiments where we compare SSM with iSSM, we use the exact same model, i.e. linear dynamics and nonlinear emissions. The only difference between SSM and iSSM is the interventional formulation. In addition, the focus of our experiments are identifiability as opposed to mere reconstruction of the data. That said, one can develop new iSSMs with different types of dynamics and emissions using the same framework, such as nonlinear dynamics and nonlinear emissions. Notice that the our inference method will still be effective in other scenarios, and only the generative model should account for the new components. This work attempts to lay the foundation of applying interventional models to neural datasets. While our theoretical results currently do not support nonlinear dynamics we’ll consider developing theoretical results for more complex models in the future work.
>
> Thanks again for your time; we look forward to your final assessment.
>
> &nbsp;
>
> ### **Further responses to reviewer xGnX**
>
> **Question 1**: Our main intuition is that the perturbations kick the state of the system outside of the "observational manifold" and force the system to explore regions in the state space that are not visited in the observational regime (we depict this in the schematic in Fig. 1). In addition, our theoretical results suggest that if the true generative model of the data follows our modeling assumptions, using interventional models allows for identifying all model components which theoretically leads to out of distribution generalization. However, in reality the true generative model of the data is far more complex and our model components only provide an approximation to their true counterparts.
>
> **Question 2:**  iSSM is equivalent to SSM in the absence of interventional data. If $\boldsymbol{u}_t = 0$ then the term that involves $\boldsymbol{u}_t$ is ignored and we recover the observational SSM model. It’s only in the presence of interventional inputs where the latent nodes dissociate from their parents in the graphical model that enable all the benefits that we exploited (enabling the model to visit new states, changing the graphical model of the data providing more information about the model, etc.).
>
> **Question 3:** Yes. Using LSTMs as the variational family has become standard in the literature of SSMs. Some benefits of LSTMs include (1) it’s causal, therefore it respects the arrow of time (2) it’s context dependent, therefore it integrates information from far away time points to inform the model predictions (3) it’s data efficient and scales well with dimensionality and time. Due to these benefits and specifically since we observed sufficiently good results we did not experiment with alternatives. The only alternative we tried was vanilla RNNs which drastically underperformed LSTMs.

---

> > ### Comment · Reviewer_3PQB · 2025-04-05
> >
> > I thank the authors for their rebuttal. I'll keep my recommendation score.

---

> > > ### Author Response · Authors · 2025-04-08
> > >
> > > Thank you for your time, please feel free to check our new results included in response to reviewer **xGnX**. We hope our new results improve the quality of the paper.

---

### Decision · Program_Chairs · 2025-05-01

**Decision:**

Accept (poster)

**Comment:**

This paper introduces the interventional state space model (iSSM), a principled extension of SSMs designed to handle external interventions in neural dynamics. The key strengths of the submission lie in its clear alignment between problem formulation, proposed method, and evaluation. Theoretical results establishing identifiability under appropriate assumptions are compelling, and the paper provides experimental evaluation on both synthetic and real-world datasets.

**However, several issues limit the paper’s overall strength**. The evaluation could be improved: the paper does not include benchmarks against other models (e.g., even a simple LDS). The comparison would have been helpful, especially since the models' performance on the simple synthetic datasets is concerning (e.g., when the number of interventions is 0, neural reconstruction is close to 0).

Their main methods figure remains unclear: the dynamics equation -- particularly in the Dynamic Attactor example -- does not appear to be consistent with the authors' figure depicting the flow field, which shows the ring attractors. Inferred flow fields were not presented initially and remain only partially analyzed post-rebuttal, which must be in the paper. There are also concerns about the choice of evaluation metrics for Figure 4 (e.g., Pearson’s r vs. log-likelihood for spike trains), and incomplete clarity around inference details despite an appendix update.

In sum, the contribution is promising with several reviewers recommending accept, yet key issues do remain. In summary, the authors should strengthen the empirical analysis, clarify model-inference details (and code), and improve quantitative evaluation of learned dynamics prior to publication.